# ELIMINATING OVERSATURATION AND ARTIFACTS OF HIGH GUIDANCE SCALES IN DIFFUSION MODELS

**Seyedmorteza Sadat[1], Otmar Hilliges[1], Romann M. Weber[2]**
[1]ETH Zürich, [2]DisneyResearch|Studios
{seyedmorteza.sadat,otmar.hilliges}@inf.ethz.ch
{romann.weber}@disneyresearch.com

## ABSTRACT

Classifier-free guidance (CFG) is crucial for improving both generation quality and alignment between the input condition and final output in diffusion models. While a high guidance scale is generally required to enhance these aspects, it also causes oversaturation and unrealistic artifacts. In this paper, we revisit the CFG update rule and introduce modifications to address this issue. We first decompose the update term in CFG into parallel and orthogonal components with respect to the conditional model prediction and observe that the parallel component primarily causes oversaturation, while the orthogonal component enhances image quality. Accordingly, we propose down-weighting the parallel component to achieve high-quality generations without oversaturation. Additionally, we draw a connection between CFG and gradient ascent and introduce a new rescaling and momentum method for the CFG update rule based on this insight. Our approach, termed adaptive projected guidance (APG), retains the quality-boosting advantages of CFG while enabling the use of higher guidance scales without oversaturation. APG is easy to implement and introduces practically no additional computational overhead to the sampling process. Through extensive experiments, we demonstrate that APG is compatible with various conditional diffusion models and samplers, leading to improved FID, recall, and saturation scores while maintaining precision comparable to CFG, making our method a superior plug-and-play alternative to standard classifier-free guidance.[1]

CFG  APG (Ours)

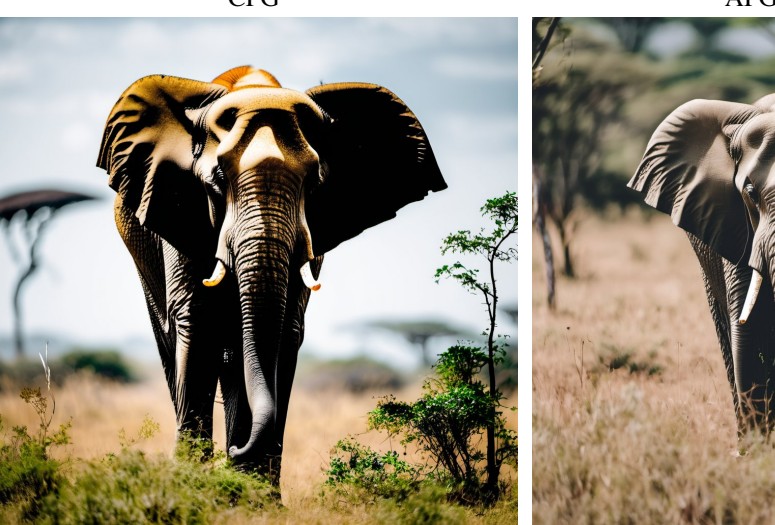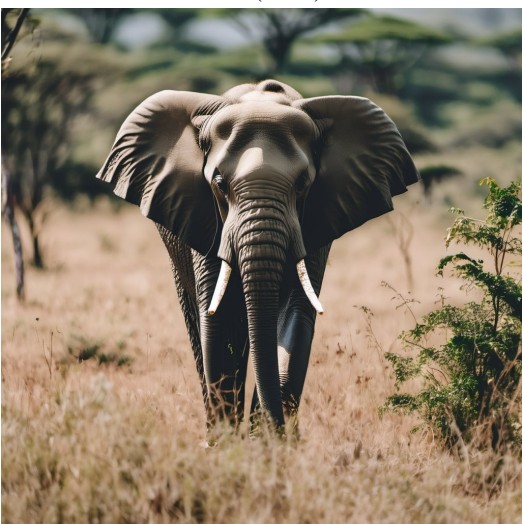

Figure 1: Classifier-free guidance is essential for generating high-quality images but causes oversaturation and unrealistic artifacts in the outputs. We introduce APG, a novel method that keeps the quality of CFG but significantly reduces its harmful oversaturation. Both images are generated with Stable Diffusion XL (Podell et al., 2023) and a guidance scale of 15.

---

[1]All visual results in the paper are best viewed in color and when zoomed in.

# 1 INTRODUCTION

Diffusion models (Sohl-Dickstein et al., 2015; Ho et al., 2020; Song et al., 2021b) are a class of generative models that learn the data distribution by reversing a forward process that adds noise to the data until the samples are indistinguishable from pure noise. Although the theory suggests that simulating the backward process in diffusion models should result in correct sampling from the data distribution, unguided sampling from diffusion models often results in low-quality images that do not align well with the input condition. Accordingly, classifier-free guidance (Ho & Salimans, 2022) has been established as an essential tool in modern diffusion models for increasing the quality of generations and the alignment between the condition and the generated image, albeit at the cost of reduced diversity (Ho & Salimans, 2022; Sadat et al., 2024a).

Modern text-to-image models, such as Stable Diffusion (Rombach et al., 2022), generally require high guidance scales in order for the generations to have better quality and align well with the input prompt. However, high guidance scales often result in oversaturated colors and simplified image compositions (Saharia et al., 2022b; Kynkäänniemi et al., 2024). Despite these disadvantages, high CFG scales are still used in practice due to their superior image quality compared to alternatives.

In this paper, we analyze the update rule of CFG and show that with a few modifications to how the CFG update is applied at inference, we can vastly mitigate the oversaturation and artifacts of high guidance scales. First, we show that the CFG update rule can be decomposed into two components, one that is parallel to the conditional model prediction, and one that is orthogonal to this prediction. We show that the orthogonal element is mainly responsible for improving image quality, while the parallel part primarily adds contrast and saturation to the output. To the best of our knowledge, this is the first study that disentangles these two effects in CFG.

Additionally, we establish a connection between the CFG update rule and stochastic gradient ascent. This insight leads us to explore a rescaled version of the CFG update direction and incorporate a momentum term, similar to adaptive optimization methods. The rescaling is motivated by the need to control large update norms, which can cause significant drifts in the sampling process. To prevent this, we constrain the updates to lie within a sphere. For the momentum term, unlike with traditional optimization, we apply a *negative* value to introduce a repulsive effect between consecutive updates, effectively down-weighting components already present in previous steps. We refer to this as *reverse momentum*. By combining rescaling, reverse momentum, and projection, we introduce a new method, called adaptive projected guidance (APG), which allows the use of higher guidance scales without oversaturation or degradation in image quality.

Through extensive experiments with several diffusion models, such as EDM2 (Karras et al., 2023) and Stable Diffusion (Rombach et al., 2022), we demonstrate that APG can utilize high guidance scales without encountering oversaturation. As a result, we conclude that APG significantly expands the usable guidance range in practice and mitigates the harmful effects of CFG at high guidance scales. Our quantitative analysis shows that replacing CFG with APG improves FID, recall, and saturation scores while maintaining precision similar to CFG. Furthermore, when combined with Stable Diffusion 3 (Esser et al., 2024), APG enhances the consistency of text rendering in generated images. We also demonstrate that APG is compatible with distilled models that use fewer sampling steps, such as SDXL-Lightning (Lin et al., 2024b). A representative visual comparison between CFG and APG is shown in Figure 1.

# 2 RELATED WORK

Score-based diffusion models (Song & Ermon, 2019; Song et al., 2021b; Sohl-Dickstein et al., 2015; Ho et al., 2020) learn data distributions by reversing a forward diffusion process that gradually corrupts data into Gaussian noise. These models have rapidly outperformed previous generative modeling methods in terms of fidelity and diversity (Nichol & Dhariwal, 2021; Dhariwal & Nichol, 2021), setting new benchmarks across various domains. They have achieved state-of-the-art results in unconditional image generation (Dhariwal & Nichol, 2021; Karras et al., 2022), text-to-image generation (Ramesh et al., 2022; Saharia et al., 2022b; Balaji et al., 2022; Rombach et al., 2022; Podell et al., 2023; Yu et al., 2022), video generation (Blattmann et al., 2023b;a; Gupta et al., 2023), image-to-image translation (Saharia et al., 2022a; Liu et al., 2023a), and audio generation (Chen et al., 2021; Kong et al., 2021; Huang et al., 2023).

Since the introduction of the DDPM model (Ho et al., 2020), numerous advancements have been made, such as improved network architectures (Hoogeboom et al., 2023; Karras et al., 2023; Peebles & Xie, 2022; Dhariwal & Nichol, 2021), enhanced sampling algorithms (Song et al., 2021a; Karras et al., 2022; Liu et al., 2022b; Lu et al., 2022a; Salimans & Ho, 2022), and new training techniques (Nichol & Dhariwal, 2021; Karras et al., 2022; Song et al., 2021b; Salimans & Ho, 2022; Rombach et al., 2022). Despite these advancements, diffusion guidance, including both classifier and classifier-free guidance (Dhariwal & Nichol, 2021; Ho & Salimans, 2022), remains crucial in enhancing generation quality and improving alignment between the condition and the output image (Nichol et al., 2022), albeit at the cost of reduced diversity and oversaturated outputs.

A recent line of work, such as CADS (Sadat et al., 2024a) and interval guidance (IG) (Kynkäänniemi et al., 2024), has focused on enhancing the diversity of generations at higher guidance scales. In contrast, our proposed method, APG, specifically addresses the oversaturation issue in CFG, as these diversity-boosting methods still struggle with oversaturation at higher guidance scales. In Appendix C.1, we demonstrate that APG can be combined with CADS and IG to achieve diverse generations without encountering oversaturation problems.

Dynamic thresholding (Saharia et al., 2022b) was introduced to mitigate the saturation effect in CFG, but it is not directly applicable to latent diffusion models (since it assumes pixel values are between $[-1, 1]$) and tends to produce images lacking in detail. Another approach, CFG Rescale (Lin et al., 2024a), aims to reduce overexposure in generated images by rescaling the standard deviation of the predictions after applying CFG. However, we demonstrate that our method is noticeably more effective at reducing oversaturation compared to CFG Rescale.

Orthogonal projection has been explored in the context of text-to-3D generation (Armandpour et al., 2023) and non-linear guidance (Zheng & Lan, 2024), but none of these methods tackle the saturation issue at higher guidance scales. We also demonstrate that naive projection has minimal impact on CFG behavior, as it must be applied to the denoised predictions to be effective. Additionally, we incorporate rescaling and reverse momentum to further mitigate the adverse effects of CFG at higher guidance scales. We show that APG can be applied to various conditional diffusion models while adding practically no overhead to the sampling process.

## 3 BACKGROUND

We provide a brief overview of diffusion models in this section. Let $x \sim p_{\text{data}}(x)$ represent a data point, and let $z_t = x + \sigma(t)\epsilon$ describe a forward process of the diffusion model that introduces noise to the data, where $t \in [0, 1]$ is the time step. Here, $\sigma(t)$ is the noise schedule, which determines the amount of information destroyed at each time step $t$, with $\sigma(0) = 0$ and $\sigma(1) = \sigma_{\max}$. Karras et al. (2022) demonstrated that this forward process is equivalent to the following ordinary differential equation (ODE):

$$\mathrm{d}z_t = -\dot{\sigma}(t)\sigma(t)\,\nabla_{z_t} \log p_t(z_t)\mathrm{d}t, \tag{1}$$

where $p_t(z_t)$ denotes the time-dependent distribution of noisy samples, with $p_0 = p_{\text{data}}$ and $p_1 = \mathcal{N}(0, \sigma_{\max}^2 \mathbf{I})$. With access to the time-dependent score function $\nabla_{z_t} \log p_t(z_t)$, one can sample from the data distribution $p_{\text{data}}$ by solving the ODE backward in time (from $t = 1$ to $t = 0$). The unknown score function $\nabla_{z_t} \log p_t(z_t)$ is estimated using a neural denoiser $D_{\theta}(z_t, t)$, which is trained to predict the clean samples $x$ from the corresponding noisy samples $z_t$. This framework also allows for conditional generation by training a denoiser $D_{\theta}(z_t, t, y)$ that incorporates additional input signals $y$, such as class labels or text prompts.

**Classifier-free guidance (CFG)** CFG is an inference method designed to enhance the quality of generated outputs by combining the predictions of a conditional model and an unconditional model (Ho & Salimans, 2022). Given a null condition $y_{\text{null}} = \varnothing$ for the unconditional case, CFG modifies the denoiser's output at each sampling step as follows:

$$\hat{D}_{\text{CFG}}(z_t, t, y) = D_{\theta}(z_t, t, y_{\text{null}}) + w(D_{\theta}(z_t, t, y) - D_{\theta}(z_t, t, y_{\text{null}})), \tag{2}$$

where $w = 1$ represents the non-guided case. The unconditional model $D_{\theta}(z_t, t, y_{\text{null}})$ is trained by randomly applying the null condition $y_{\text{null}} = \varnothing$ to the denoiser's input for a portion of training. Alternatively, a separate denoiser can be trained to estimate the unconditional score in Equation (2) (Karras et al., 2023). Similar to the truncation method used in GANs (Brock et al., 2019), CFG improves the quality of images but reduces diversity (Murphy, 2023).

## 4  ADAPTIVE PROJECTED GUIDANCE

We now present our method for addressing oversaturation and artifacts in CFG at high guidance scales. Let $\Delta D_t = D_{\boldsymbol{\theta}}(\boldsymbol{z}_t, t, \boldsymbol{y}) - D_{\boldsymbol{\theta}}(\boldsymbol{z}_t, t, \boldsymbol{y}_{\text{null}})$ be the CFG update direction at time step $t$. Note that Equation (2) can now be written as

$$\hat{D}_{\text{CFG}}(\boldsymbol{z}_t, t, \boldsymbol{y}) = D_{\boldsymbol{\theta}}(\boldsymbol{z}_t, t, \boldsymbol{y}) + (w - 1)\Delta D_t. \tag{3}$$

(See Appendix A for the derivation.) We use Equation (3) for the rest of this paper to motivate our changes. APG has three elements: (1) projection, (2) rescaling, and (3) reverse momentum. We discuss each component below.

**Orthogonal projection**   First, note that we can decompose $\Delta D_t$ into two different components: $\Delta D_t^{\parallel}$, which is parallel to $D_{\boldsymbol{\theta}}(\boldsymbol{z}_t, t, \boldsymbol{y})$, and $\Delta D_t^{\perp}$, which is orthogonal to $D_{\boldsymbol{\theta}}(\boldsymbol{z}_t, t, \boldsymbol{y})$, i.e., $\Delta D_t = \Delta D_t^{\perp} + \Delta D_t^{\parallel}$. We can compute $\Delta D_t^{\parallel}$ via orthogonal projection, with

$$\Delta D_t^{\parallel} = \frac{\langle \Delta D_t, D_{\boldsymbol{\theta}}(\boldsymbol{z}_t, t, \boldsymbol{y}) \rangle}{\langle D_{\boldsymbol{\theta}}(\boldsymbol{z}_t, t, \boldsymbol{y}), D_{\boldsymbol{\theta}}(\boldsymbol{z}_t, t, \boldsymbol{y}) \rangle} D_{\boldsymbol{\theta}}(\boldsymbol{z}_t, t, \boldsymbol{y}). \tag{4}$$

We then have $\Delta D_t^{\perp} = \Delta D_t - \Delta D_t^{\parallel}$. We observe that the orthogonal component is chiefly responsible for improvements in image quality, while the parallel component increases saturation in the generations as shown in Figure 2.

Accordingly, we modify the update direction to form $\Delta D_t(\eta) = \Delta D_t^{\perp} + \eta \Delta D_t^{\parallel}$, where $\eta \leq 1$ is a hyperparameter. Note that $\Delta D_t(1)$ is identical to the unmodified CFG update direction described above. We show that reducing the strength of the parallel component (i.e. setting $\eta$ close to zero) significantly reduces saturation and results in more realistic generations at higher guidance scales.

The intuition behind the saturating effect of the parallel component is helped by thinking of the output $D_{\boldsymbol{\theta}}(\boldsymbol{z}_t, t, \boldsymbol{y})$ as an image with a typical range of values.[2] When an update parallel to this image is added, it serves to create a "gain," pushing the values toward the extremes of their range. This gain effect can be seen by direct calculation:

$$D_{\boldsymbol{\theta}}(\boldsymbol{z}_t, t, \boldsymbol{y}) + (w - 1)\Delta D_t^{\parallel} = \left[ 1 + (w - 1)\frac{\|\Delta D_t^{\parallel}\|}{\|D_{\boldsymbol{\theta}}(\boldsymbol{z}_t, t, \boldsymbol{y})\|} \right] D_{\boldsymbol{\theta}}(\boldsymbol{z}_t, t, \boldsymbol{y}), \tag{5}$$

where we note that the term in brackets on the right-hand side is greater than one for $w > 1$. Thus, this term only adds saturation to the predictions $D_{\boldsymbol{\theta}}(\boldsymbol{z}_t, t, \boldsymbol{y})$ during each inference step, much like multiplying pixel values by a number greater than one. We show in Section 5.2 that reducing $\eta$ and leaning more heavily on the orthogonal component significantly attenuates this saturation side effect in generations while maintaining the quality-boosting benefits of CFG.

**Adding rescaling**   Next, we argue that the CFG update rule in Equation (3) can be interpreted as one step of gradient ascent on the $\ell_2$ distance between the conditional and unconditional prediction, i.e., one step of gradient ascent on $\frac{1}{2}\|D_{\boldsymbol{\theta}}(\boldsymbol{z}_t, t, \boldsymbol{y}) - D_{\boldsymbol{\theta}}(\boldsymbol{z}_t, t, \boldsymbol{y}_{\text{null}})\|^2$ with a learning rate of $w - 1$. (See Appendix A for proof.) Inspired by this interpretation and normalized gradient ascent, we rescale the CFG update rule at each step to regulate the impact of each update. Specifically, we constrain $\Delta D_t$ to be inside a sphere with radius $r$ via

$$\Delta D_t \leftarrow \Delta D_t \cdot \min\left( 1, \frac{r}{\|\Delta D_t\|} \right), \tag{6}$$

where $r$ is a hyperparameter. This rescaling ensures that the CFG update $\Delta D_t$ stays closer to $D_{\boldsymbol{\theta}}(\boldsymbol{z}_t, t, \boldsymbol{y})$, limiting drift at each sampling step if $\|\Delta D_t\|$ is large. As demonstrated in Section 5.2, this adjustment improves both FID and recall.

**Adding reverse momentum**   Finally, leveraging the connection to gradient ascent, we introduce a reverse momentum term to the CFG update rule. We define the momentum for the CFG update direction as $\overline{\Delta D_t} \leftarrow \Delta D_t + \beta \overline{\Delta D_t}$, where $\overline{\Delta D_t} = 0$ initially. The momentum term accounts for the average values of past updates; however, unlike standard optimization methods, we use a *negative*

---

[2]This intuition also holds for the image-like representations in latent diffusion models.

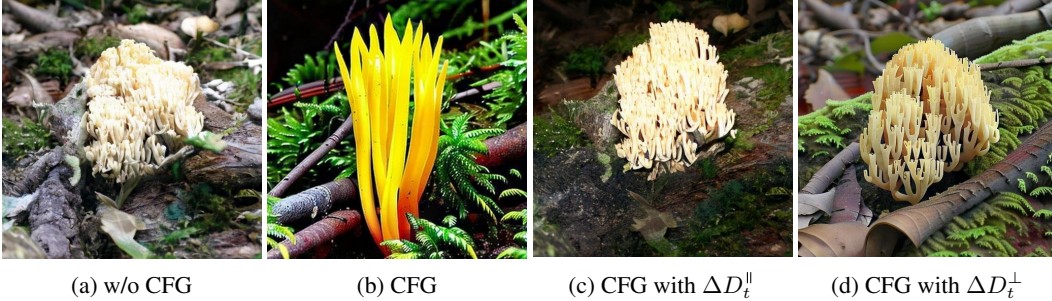

| (a) w/o CFG | (b) CFG | (c) CFG with $\Delta D_t^{\parallel}$ | (d) CFG with $\Delta D_t^{\perp}$ |

Figure 2: Influence of the parallel and orthogonal components ($\Delta D_t^{\parallel}$ and $\Delta D_t^{\perp}$) in CFG. (a) The generation without CFG lacks quality and detail. (b) Applying CFG increases quality but introduces oversaturation. (c) Applying CFG only with the parallel component $\Delta D_t^{\parallel}$ barely changes the output quality compared to (a) and only increases saturation. (d) Applying CFG with only the orthogonal part $\Delta D_t^{\perp}$ enhances image quality without causing oversaturation.

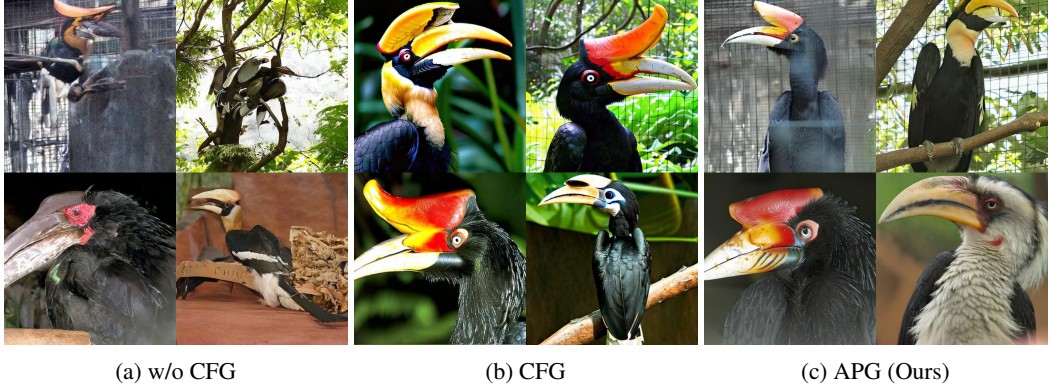

| (a) w/o CFG | (b) CFG | (c) APG (Ours) |

Figure 3: Illustrating the effect of APG on generated images . (a) Sampling without guidance leads to low-quality generations. (b) CFG improves image quality but causes oversaturation. (c) Using APG instead of CFG results in high-quality generations without oversaturation.

momentum strength $\beta < 0$. Intuitively, this pushes the model away from previous CFG update directions and encourages the model to focus more on the current update direction. As shown in Section 5.2, incorporating reverse momentum further enhances image quality (i.e., lower FID scores).

APG is easy to implement, and we provide the source code in Algorithm 1 (appendix). As shown in Section 5.2, it is crucial to convert the diffusion model's outputs (e.g., predicted noise) into the denoised prediction $D_{\boldsymbol{\theta}}(\boldsymbol{z}_t, t, \boldsymbol{y})$ in order to perform the projection. Further details on obtaining $D_{\boldsymbol{\theta}}(\boldsymbol{z}_t, t, \boldsymbol{y})$ for common prediction types are discussed in Appendix B. Figure 3 demonstrates that using APG instead of CFG produces high-quality generations without oversaturation or the undesirable artifacts associated with high guidance scales.

## 5 EXPERIMENTS AND RESULTS

**Setup** We mainly experiment with text-to-image generation with Stable Diffusion (Rombach et al., 2022) and class-conditional ImageNet (Russakovsky et al., 2015) generation using EDM2 (Karras et al., 2023) and DiT-XL/2 (Peebles & Xie, 2022). For all experiments, we use the default diffusion sampler from each model (e.g., Euler scheduler for Stable Diffusion XL) along with pretrained checkpoints and corresponding codebases to ensure consistency in weights and the sampling process with the original frameworks.

**Distribution metrics** We use Fréchet Inception Distance (FID) (Heusel et al., 2017) as our primary metric for evaluating both the quality and diversity of generated images due to its alignment with human judgment. Since FID is sensitive to small implementation details, we ensure that all models are evaluated under the same setup. For completeness, we also report precision (Kynkäänniemi et al., 2019) as an additional quality metric and recall (Kynkäänniemi et al., 2019) as a diversity metric.

| CFG | APG (Ours) | CFG | APG (Ours) |
| --- | --- | --- | --- |

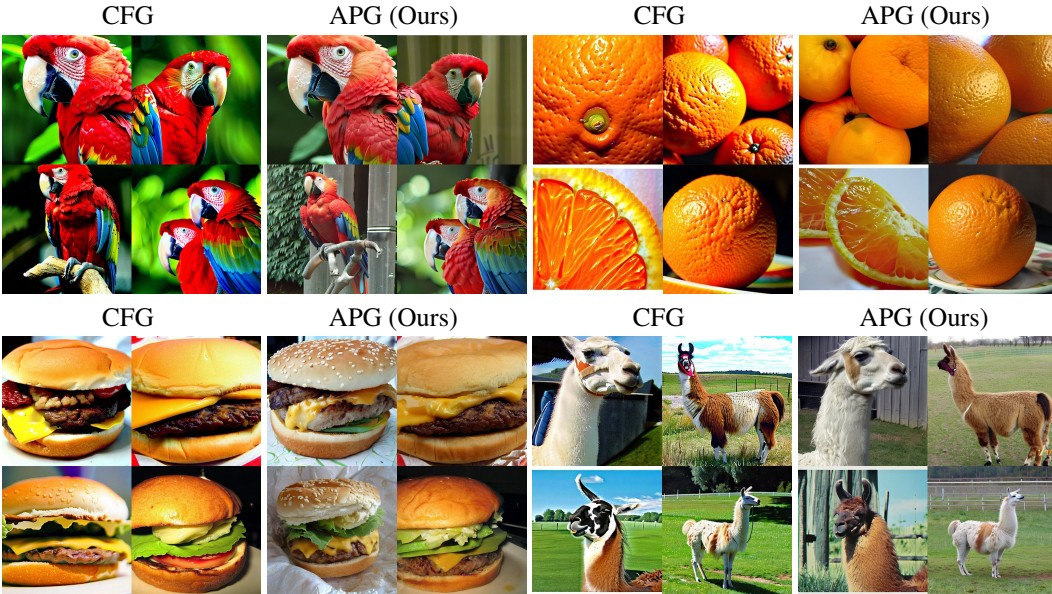

Figure 4: Class-conditional generation results using EDM2 with $w = 4$. APG significantly reduces saturation in the generations while keeping the high-quality global structure of each image.

| CFG | APG (Ours) | CFG | APG (Ours) |
| --- | --- | --- | --- |

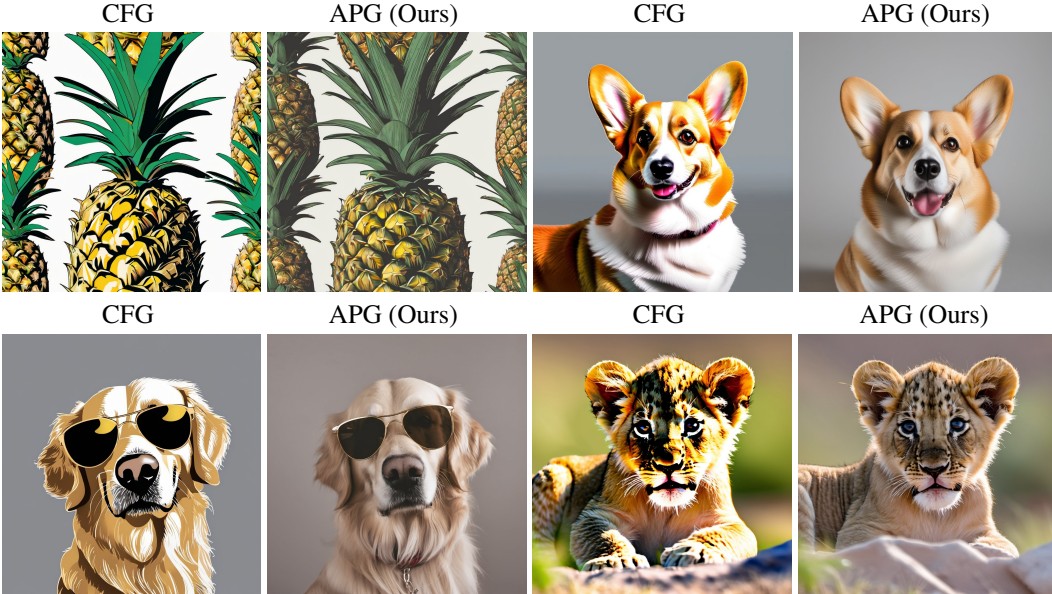

Figure 5: Text-to-image generation results using Stable Diffusion XL with $w = 15$. APG produces more realistic images compared to the oversaturated outputs of CFG.

**Color metrics** While FID measures the overall quality of generated images, we introduce specific metrics to directly assess saturation and contrast. To measure saturation, we convert each image from RGB to HSV and compute the mean of the saturation channel. We define contrast (also known as RMS contrast) as the standard deviation of pixel values after converting the image to grayscale. The final metrics are derived by averaging the saturation and contrast values across all generated images.

## 5.1 MAIN RESULTS

**Qualitative results** Figures 4 and 5 present our qualitative results comparing APG with CFG for EDM2 and Stable Diffusion XL. We observe that, compared to CFG, APG generates more realistic images with noticeably lower saturation. Furthermore, APG appears to produce fewer artifacts in the final outputs, as illustrated in Figure 6. Additional visual results can be found in Appendix E.

| CFG | APG (Ours) | CFG | APG (Ours) |

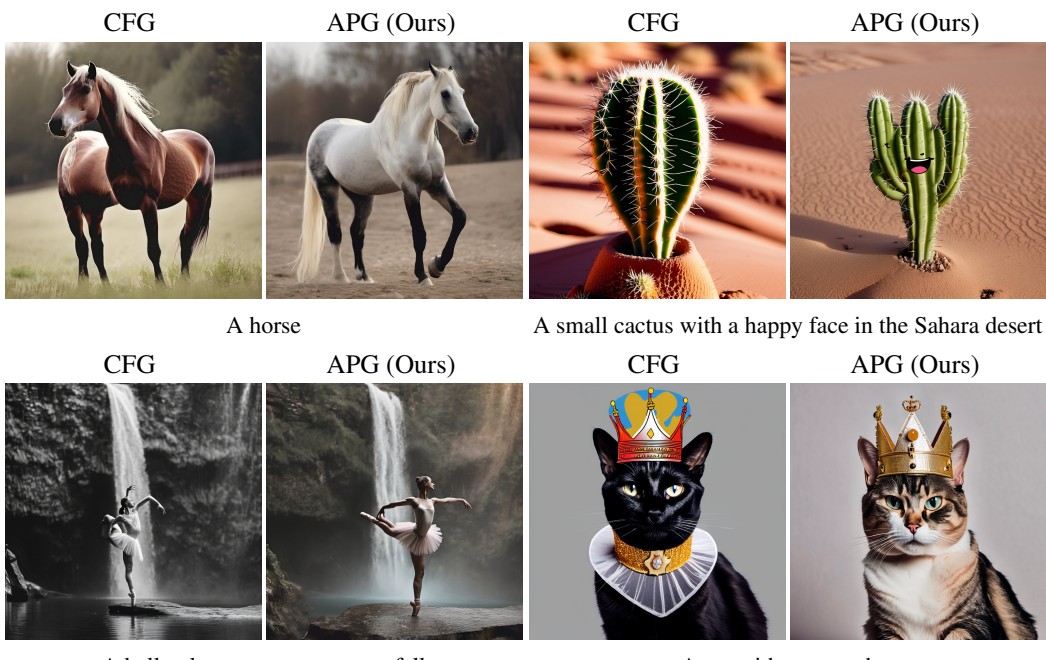

A horse  A small cactus with a happy face in the Sahara desert

| CFG | APG (Ours) | CFG | APG (Ours) |

A ballet dancer next to a waterfall  A cat with a queen hat

Figure 6: Examples of artifacts in the CFG outputs that can be solved by using APG. We see that for all images, APG outputs follow the prompt with a more globally consistent generation.

Table 1: Quantitative comparison between CFG and APG. APG consistently improves FID, recall and color metrics while maintaining similar or better precision compared to CFG.

| Model | Guidance | FID ↓ | Precision ↑ | Recall ↑ | Saturation ↓ | Contrast ↓ |
|---|---|---|---|---|---|---|
| EDM2-S ($w = 4$) | CFG | 10.42 | **0.85** | 0.48 | 0.46 | 0.27 |
| | APG (Ours) | **6.49** | **0.85** | **0.62** | **0.33** | **0.21** |
| EDM2-XXL ($w = 2$) | CFG | 8.65 | **0.84** | 0.57 | 0.37 | 0.23 |
| | APG (Ours) | **4.94** | 0.83 | **0.67** | **0.31** | **0.21** |
| DiT-XL/2 ($w = 4$) | CFG | 19.14 | **0.92** | 0.35 | 0.37 | 0.25 |
| | APG (Ours) | **9.34** | 0.89 | **0.56** | **0.30** | **0.20** |
| Stable Diffusion 2.1 ($w = 10$) | CFG | 27.53 | 0.65 | 0.41 | 0.36 | 0.27 |
| | APG (Ours) | **22.21** | **0.67** | **0.49** | **0.27** | **0.22** |
| Stable Diffusion XL ($w = 15$) | CFG | 26.29 | 0.62 | 0.49 | 0.28 | 0.24 |
| | APG (Ours) | **25.35** | **0.64** | **0.50** | **0.18** | **0.17** |

**Quantitative results** We next present a quantitative comparison between APG and CFG in Table 1. The table shows that APG outperforms CFG across multiple models, consistently achieving better FID and recall scores, as well as lower saturation and contrast. Moreover, APG demonstrates similar precision to CFG, indicating that the reduction in saturation does not compromise the quality of individual samples.

**Distribution of pixel values** Figure 7 presents the kernel density estimate (KDE) plot of RGB and saturation values for 100 images generated using CFG and APG, along with KDE plots for 100 real samples drawn from the evaluation subset of ImageNet. Compared to CFG, APG plots are more broadly distributed across the spectrum with less concentration at the extremes. This indicates that images generated with APG are closer to real data in terms of saturation and color composition.

**APG vs guidance scale** In Figure 8, we demonstrate that as the guidance scale increases, APG consistently achieves lower FID and higher recall while maintaining similar or better precision compared to CFG. Additionally, CFG exhibits increasing saturation at higher guidance scales, whereas APG maintains a relatively constant saturation level. Therefore, APG allows the usage of higher guidance scales, achieving better FID and diversity without oversaturation.

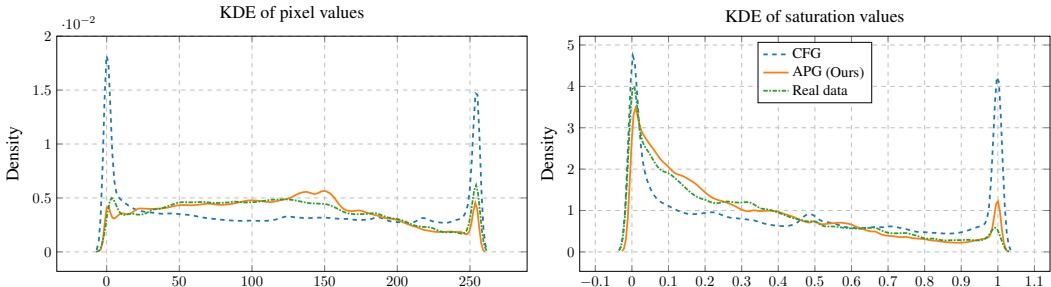

Figure 7: Kernel density estimates of pixel and saturation values for two sets of samples generated with CFG and APG. Compared to CFG, images generated with APG show less concentration around saturated pixels, indicated by the spikes at the extreme values in both plots.

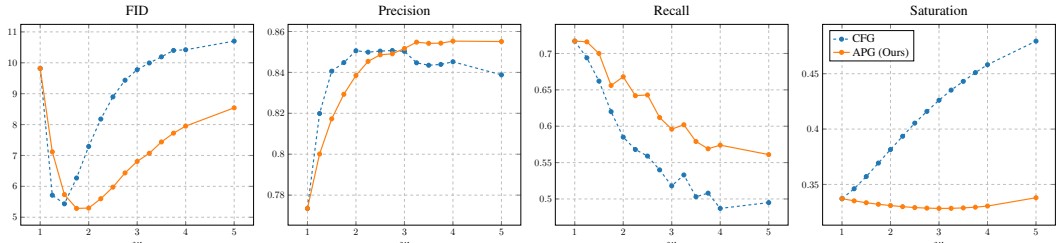

Figure 8: Comparison between CFG and APG as the guidance scale increases. APG offers better FID and recall while maintaining similar or better precision to CFG at higher guidance scales.

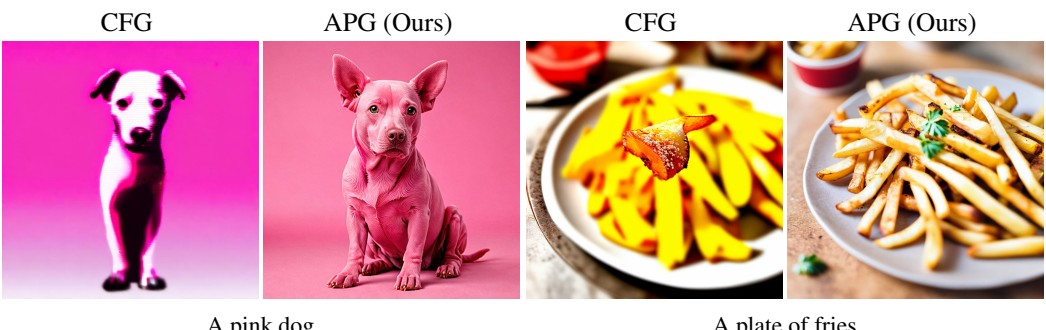

Figure 9: Showcasing the compatibility of APG with distilled diffusion models using SDXL-Lightning. Compared to CFG, using APG does not result in degradation in the output quality.

**Improving diversity**  While APG is designed to address oversaturation at high guidance scales, we also observed that it can enhance the diversity of generations. As shown in Table 1 and Figure 8, APG improves distribution coverage (i.e., higher recall) while maintaining precision comparable to CFG. Additional qualitative results illustrating the enhanced diversity are provided in Figure 17 (appendix).

**Using APG with distilled models**  A common issue with CFG is that it degrades the quality of final outputs when applied to distilled models with fewer sampling steps (e.g., 8-step SDXL-Lightning (Lin et al., 2024c)). In this section, we show that APG does not encounter this problem and can be effectively applied to distilled models. Figure 9 demonstrates that replacing CFG with APG significantly improves generation quality. Extended results with additional models are provided in Appendix C.3, along with more visual examples in Appendix E.

**Text spelling with Stable Diffusion 3**  Next, we demonstrate that integrating APG with Stable Diffusion 3 (Esser et al., 2024) enhances the consistency of text rendering in generated images. As shown in Figure 10, APG produces more accurate spelling in the generated images compared to standard CFG. More visual results are given in Figure 20 (appendix).

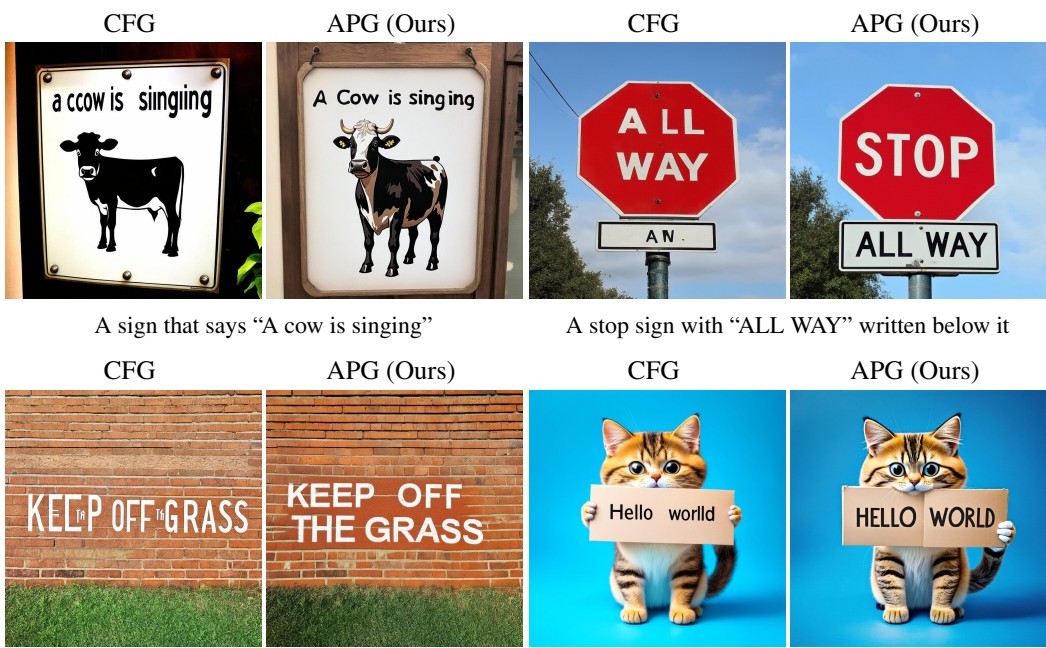

Figure 10: Comparison of CFG and APG for text quality in generated images using Stable Diffusion 3 (Esser et al., 2024). In contrast to CFG, APG consistently produces correct spellings.

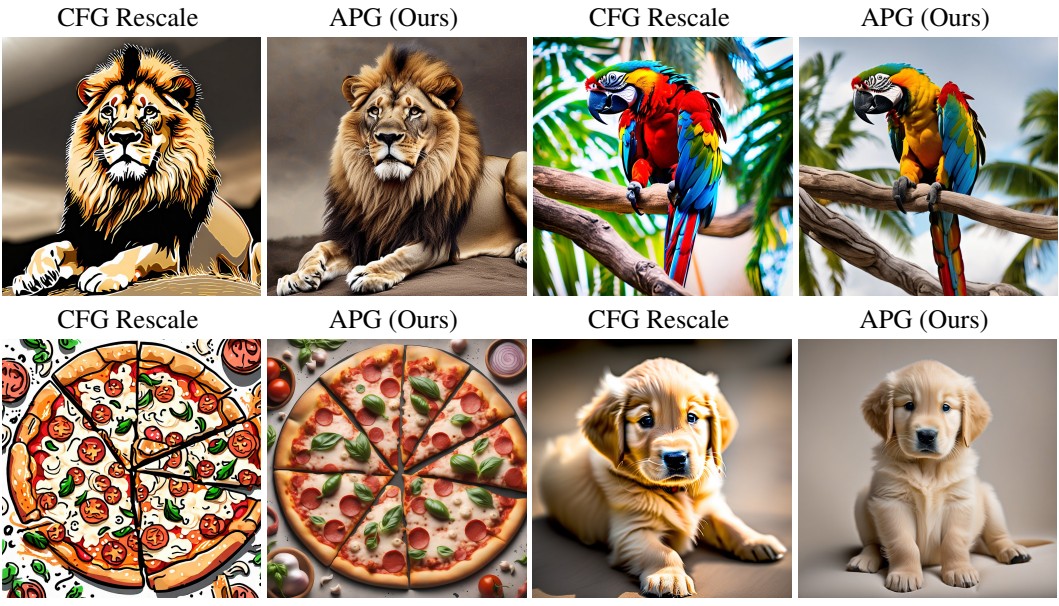

Figure 11: Comparison between APG and CFG Rescale using Stable Diffusion XL. CFG Rescale is unable to solve the saturation issue at high guidance scales compared with APG.

**Comparison with CFG Rescale**    CFG Rescale was introduced in (Lin et al., 2024a) as a method to reduce saturation at high guidance scales. In this section, we demonstrate that APG is more effective than CFG Rescale. The comparison in Figure 11 shows that APG outputs have significantly less saturation and are more realistic than those produced with CFG Rescale.

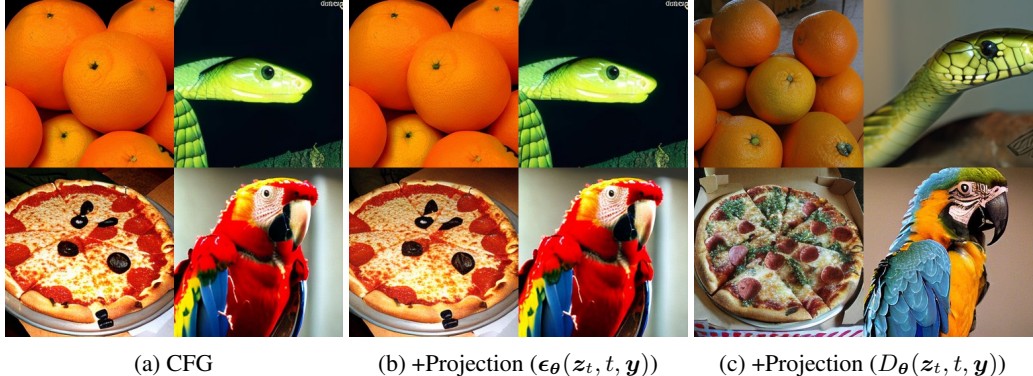

(a) CFG      (b) +Projection ($\epsilon_{\boldsymbol{\theta}}(\boldsymbol{z}_t, t, \boldsymbol{y})$)      (c) +Projection ($D_{\boldsymbol{\theta}}(\boldsymbol{z}_t, t, \boldsymbol{y})$)

Figure 12: The importance of projecting onto the denoised samples. When performing projection w.r.t. the predicted noise (b), the outputs are barely different than standard CFG (a). However, projecting onto denoised samples (c) more effectively reduces saturation.

**Computational cost of APG** The computational cost of APG is practically identical to that of CFG, as the rescaling and projection steps incur negligible overhead compared to querying the denoiser. Specifically, in the case of Stable Diffusion XL, the forward pass through the diffusion network takes approximately 130 milliseconds on an RTX 3090 GPU for a single image, while the guidance step requires only about 0.45 milliseconds.

## 5.2 ABLATION STUDIES

We now present our ablation studies in this section. The experiments are based on class-conditional generation using the EDM2 model (Karras et al., 2023), with FID as the primary metric to justify our design choices. First, Table 2 highlights the importance of each component in APG. We observe that removing projection, rescaling, or reverse momentum results in higher FID scores. Additionally, note that the projection

Table 2: Importance of different components in APG.

| Config | FID ↓ | Recall ↑ | Saturation ↓ |
|---|---|---|---|
| APG ($w = 4$) | **6.49** | **0.62** | **0.33** |
| w/o projection | 6.63 | 0.60 | 0.37 |
| w/o rescaling | 7.93 | 0.56 | 0.34 |
| w/o momentum | 6.85 | 0.61 | **0.33** |

component is primarily responsible for reducing saturation while rescaling and reverse momentum mainly improve FID and recall. Appendix C.8 gives extended ablation results on the effect of each component in APG.

**Importance of the model prediction type** While CFG works the same across all model prediction types, we observed that our method performs best when applied to the denoised predictions $D_{\boldsymbol{\theta}}(\boldsymbol{z}_t, t, \boldsymbol{y})$, rather than, for example, the noise prediction $\epsilon_{\boldsymbol{\theta}}(\boldsymbol{z}_t, t, \boldsymbol{y})$. This is illustrated in Figure 12, where projecting onto $\epsilon_{\boldsymbol{\theta}}(\boldsymbol{z}_t, t, \boldsymbol{y})$ produces results nearly identical to CFG, while projecting onto $D_{\boldsymbol{\theta}}(\boldsymbol{z}_t, t, \boldsymbol{y})$ significantly reduces saturation. Note that as discussed in Appendix B, this is not a bottleneck for APG as various prediction types can be readily converted to $D_{\boldsymbol{\theta}}(\boldsymbol{z}_t, t, \boldsymbol{y})$ at each step.

## 6 CONCLUSION AND DISCUSSION

In this work, we investigated the oversaturation effect of high CFG scales and introduced a new method, adaptive projected guidance (APG), that achieves the same quality-boosting benefits as CFG without causing oversaturation. The key idea behind APG is to project the CFG update onto the denoised prediction of the diffusion model $D_{\boldsymbol{\theta}}(\boldsymbol{z}_t, t, \boldsymbol{y})$ and remove or down-weight the component parallel to that prediction. Additionally, by linking CFG to gradient ascent, we demonstrated that its performance can be further enhanced by incorporating rescaling and reverse momentum. Through extensive experiments, we showed that APG improves FID, recall, and saturation metrics compared to CFG, while maintaining similar or better precision. Thus, APG offers a plug-and-play alternative to standard CFG capable of delivering superior results with practically no additional computational overhead. Like CFG, challenges remain in accelerating APG so that the sampling cost approaches that of the unguided sampling (i.e., removing the need to query the diffusion network twice at each sampling step). We consider this a promising direction for future research.

## ETHICS STATEMENT

As generative modeling continues to evolve, the potential to create fake or erroneous data increases. While advancements in AI-generated content can enhance efficiency and foster creativity, it is crucial to address the associated ethical concerns. For a more detailed discussion on ethics and creativity in computer vision, we recommend Rostamzadeh et al. (2021).

## REPRODUCIBILITY STATEMENT

This work builds on the official implementations of the pretrained models referenced in the main text. The source code for implementing APG is provided in Algorithm 1, and Appendix D outlines additional implementation details, including the hyperparameters used in the main experiments.

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

## A  DETAILS ON CFG AS GRADIENT ASCENT

In this section, we discuss how CFG can be interpreted as one step of gradient ascent. To begin, note that the CFG update rule can be expressed as:

$$\hat{D}_{\text{CFG}}(z_t, t, y) = D_\theta(z_t, t, y_{\text{null}}) + w(D_\theta(z_t, t, y) - D_\theta(z_t, t, y_{\text{null}})) \tag{7}$$

$$= w D_\theta(z_t, t, y) + (1 - w) D_\theta(z_t, t, y_{\text{null}}) \tag{8}$$

$$= D_\theta(z_t, t, y) + (w - 1) D_\theta(z_t, t, y) + (1 - w) D_\theta(z_t, t, y_{\text{null}}) \tag{9}$$

$$= D_\theta(z_t, t, y) + (w - 1)(D_\theta(z_t, t, y) - D_\theta(z_t, t, y_{\text{null}})) \tag{10}$$

$$= D_\theta(z_t, t, y) + \gamma \Delta D_t, \tag{11}$$

where $\gamma = w - 1$, and $\Delta D_t = D_\theta(z_t, t, y) - D_\theta(z_t, t, y_{\text{null}})$. Next, observe that we can write:

$$D_\theta(z_t, t, y) - D_\theta(z_t, t, y_{\text{null}}) = \nabla_{D_\theta(z_t, t, y)} \left[ \frac{1}{2} \| D_\theta(z_t, t, y) - D_\theta(z_t, t, y_{\text{null}}) \|^2 \right]. \tag{12}$$

Thus, if we define the CFG objective function as

$$f_{\text{CFG}}(D_\theta(z_t, t, y), D_\theta(z_t, t, y_{\text{null}})) = \frac{1}{2} \| D_\theta(z_t, t, y) - D_\theta(z_t, t, y_{\text{null}}) \|^2, \tag{13}$$

the CFG update rule becomes equivalent to:

$$\hat{D}_{\text{CFG}}(z_t, t, y) = D_\theta(z_t, t, y) + \gamma \nabla_{D_\theta(z_t, t, y)} f_{\text{CFG}}(D_\theta(z_t, t, y), D_\theta(z_t, t, y_{\text{null}})). \tag{14}$$

Hence, we have shown that the CFG update rule corresponds to a single step of gradient *ascent* with respect to the objective function $f_{\text{CFG}}(D_\theta(z_t, t, y), D_\theta(z_t, t, y_{\text{null}}))$.

This interpretation motivated us to incorporate rescaling into standard CFG. Since the objective function $f_{\text{CFG}}(D_\theta(z_t, t, y), D_\theta(z_t, t, y_{\text{null}}))$ does not have a maximum, the CFG update step may result in arbitrary drift from $D_\theta(z_t, t, y)$. By applying rescaling, we constrain the CFG update to remain within a ball of limited radius around $D_\theta(z_t, t, y)$. The reverse momentum method is similarly inspired by this interpretation, where each update is pushed away from previous predictions.

## B  DENOISED PREDICTION FOR DIFFERENT DIFFUSION MODELS

We next briefly outline the process of computing the denoised prediction $D_\theta(z_t, t, y)$ for various diffusion models. For further details, we refer readers to Kingma & Gao (2023). In the following, let $x$ represent the clean data, $y$ a condition or class, and $\epsilon \sim \mathcal{N}(0, I)$ the noise. Given a noisy sample $z_t$ at time step $t$, the objective is to recover the clean data $x$ that produced $z_t$. The denoised version of $z_t$, which approximates $x$, is estimated by a neural network, denoted as $D_\theta(z_t, t, y)$. Before applying APG, we always convert all model predictions to $D_\theta(z_t, t, y)$. This conversion is compatible with most samplers based on the denoising framework, such as EDM (Karras et al., 2022) and DPM++ (Lu et al., 2022b). The conversions for various models are derived below, and a summary is provided in Table 3.

**DDPM**  For models using the DDPM framework (Ho et al., 2020), the forward diffusion process is defined as $z_t = \alpha_t x + \sigma_t \epsilon$, where $\sigma_t^2 + \alpha_t^2 = 1$. These models typically predict the total added noise $\epsilon$ via a neural network $\epsilon_\theta(z_t, t, y)$. Given the prediction of the model, the denoised prediction can be estimated via

$$D_\theta(z_t, t, y) = \frac{z_t - \sigma_t \epsilon_\theta(z_t, t, y)}{\alpha_t}. \tag{15}$$

If the model predicts the velocity $v = \alpha_t \epsilon - \sigma_t x$, we have

$$v = \alpha_t \frac{z_t - \alpha_t x}{\sigma_t} - \sigma_t x = \frac{\alpha_t z_t - \alpha_t^2 x - \sigma_t^2 x}{\sigma_t} = \frac{\alpha_t z_t - x}{\sigma_t}. \tag{16}$$

This leads to the following formulation for the denoised prediction:

$$D_\theta(z_t, t, y) = \alpha_t z_t - \sigma_t v_\theta(z_t, t, y). \tag{17}$$

Table 3: Summary of calculating denoised predictions $D_{\boldsymbol{\theta}}(\boldsymbol{z}_t, t, \boldsymbol{y})$ for different diffusion models.

| Config | Forward process $\boldsymbol{z}_t$ | Model prediciton | Denoised prediction $D_{\boldsymbol{\theta}}(\boldsymbol{z}_t, t, \boldsymbol{y})$ |
|---|---|---|---|
| DDPM | $\alpha_t \boldsymbol{x} + \sigma_t \boldsymbol{\epsilon}$ | $\boldsymbol{\epsilon_\theta}(\boldsymbol{z}_t, t, \boldsymbol{y})$ | $(\boldsymbol{z}_t - \sigma_t \boldsymbol{\epsilon_\theta}(\boldsymbol{z}_t, t, \boldsymbol{y}))/\alpha_t$ |
| DDPM | $\alpha_t \boldsymbol{x} + \sigma_t \boldsymbol{\epsilon}$ | $\boldsymbol{v_\theta}(\boldsymbol{z}_t, t, \boldsymbol{y})$ | $\alpha_t \boldsymbol{z}_t - \sigma_t \boldsymbol{v_\theta}(\boldsymbol{z}_t, t, \boldsymbol{y})$ |
| EDM | $\boldsymbol{x} + \sigma(t)\boldsymbol{\epsilon}$ | $F_{\boldsymbol{\theta}}(c_{\text{in}}(t)\boldsymbol{z}_t, c_{\text{noise}}(t), \boldsymbol{y})$ | $c_{\text{skip}}(t)\boldsymbol{z}_t + c_{\text{out}}(t)F_{\boldsymbol{\theta}}(c_{\text{in}}(t)\boldsymbol{z}_t, c_{\text{noise}}(t), \boldsymbol{y})$ |
| Rectified flow | $(1-t)\boldsymbol{x} + t\boldsymbol{\epsilon}$ | $\boldsymbol{v_\theta}(\boldsymbol{z}_t, t, \boldsymbol{y})$ | $\boldsymbol{z}_t - t\boldsymbol{v_\theta}(\boldsymbol{z}_t, t, \boldsymbol{y})$ |

**EDM framework**  For the EDM framework (Karras et al., 2022), the forward process is described by $\boldsymbol{z}_t = \boldsymbol{x} + \sigma(t)\boldsymbol{\epsilon}$, and the denoised prediction $D_{\boldsymbol{\theta}}(\boldsymbol{z}_t, t, \boldsymbol{y})$ is formulated via

$$D_{\boldsymbol{\theta}}(\boldsymbol{z}_t, t, \boldsymbol{y}) = c_{\text{skip}}(t)\boldsymbol{z}_t + c_{\text{out}}(t)F_{\boldsymbol{\theta}}(c_{\text{in}}(t)\boldsymbol{z}_t, c_{\text{noise}}(t), \boldsymbol{y}), \tag{18}$$

where $F_{\boldsymbol{\theta}}(c_{\text{in}}(t)\boldsymbol{z}_t, c_{\text{noise}}(t), \boldsymbol{y})$ is the output of the neural network. The EDM framework uses $\sigma(t) \propto t$; thus, $\sigma$ and $t$ can be used interchangeably in this framework.

**Rectified flow models**  For rectified flow models (Liu et al., 2023b), such as Stable Diffusion 3 (Esser et al., 2024), the forward process is given by $\boldsymbol{z}_t = (1-t)\boldsymbol{x} + t\boldsymbol{\epsilon}$. The model predicts the velocity field given by $\boldsymbol{v} = \boldsymbol{\epsilon} - \boldsymbol{x}$. Accordingly, we have

$$\boldsymbol{v} = \boldsymbol{\epsilon} - \boldsymbol{x} = \frac{\boldsymbol{z}_t - (1-t)\boldsymbol{x}}{t} - \boldsymbol{x} = \frac{\boldsymbol{z}_t - (1-t)\boldsymbol{x} - t\boldsymbol{x}}{t} = \frac{\boldsymbol{z}_t - \boldsymbol{x}}{t}. \tag{19}$$

Thus, the denoised prediction can be determined by:

$$D_{\boldsymbol{\theta}}(\boldsymbol{z}_t, t, \boldsymbol{y}) = \boldsymbol{z}_t - t\boldsymbol{v_\theta}(\boldsymbol{z}_t, t, \boldsymbol{y}) = \boldsymbol{z}_t - \sigma_t \boldsymbol{v_\theta}(\boldsymbol{z}_t, t, \boldsymbol{y}), \tag{20}$$

where we define $\sigma_t = t$.

This section demonstrates the effect of applying APG on a toy example to illustrate the differences between APG and CFG. We use a mixture of two high-dimensional Gaussians as the data distribution, which allows us to analytically compute the score functions during the diffusion process, eliminating potential errors introduced by a denoiser network. Specifically, the data distribution $p_{\text{data}}$ is defined as:

$$p_{\text{data}}(\boldsymbol{x}) = \frac{1}{2}\mathcal{N}(\boldsymbol{\mu}_1, \sigma^2\boldsymbol{I})(\boldsymbol{x}) + \frac{1}{2}\mathcal{N}(\boldsymbol{\mu}_2, \sigma^2\boldsymbol{I})(\boldsymbol{x}), \tag{21}$$

where $\boldsymbol{\mu}_1 = [-2, -2, \ldots, -2]$, $\boldsymbol{\mu}_2 = [2, 2, \ldots, 2]$, and $\sigma = 0.25$. We use a dimensionality of 500 for each component. Accordingly, the conditional distributions are equal to

$$p_{\text{data}}(\boldsymbol{x}|y=1) = \mathcal{N}(\boldsymbol{\mu}_1, \sigma^2\boldsymbol{I})(\boldsymbol{x}) \quad \text{and} \quad p_{\text{data}}(\boldsymbol{x}|y=2) = \mathcal{N}(\boldsymbol{\mu}_2, \sigma^2\boldsymbol{I})(\boldsymbol{x}). \tag{22}$$

The sampling results are shown in Figure 13 (visualizing the first two dimensions of each Gaussian). When CFG is applied with a high guidance scale, it results in a drift toward regions less likely according to the data distribution. In contrast, applying APG corrects this drift and improves mode coverage. While this is a simplified example, we argue that a similar phenomenon occurs when applying CFG to images, leading to artifacts and oversaturation in the final outputs.

## C  ADDITIONAL EXPERIMENTS

Additional experiments and ablation studies are included in this section. Unless stated otherwise, the experiments are conducted using class-conditional ImageNet (Russakovsky et al., 2015) generation.

### C.1  COMPATIBILITY WITH CADS AND IG

We first demonstrate that APG is compatible with CADS (Sadat et al., 2024a) and interval guidance (IG) (Kynkäänniemi et al., 2024), both of which are designed to enhance the diversity of generations at high guidance scales. The results, shown in Table 4, indicate that replacing CFG with APG leads to improved FID, recall, and saturation scores for both methods.

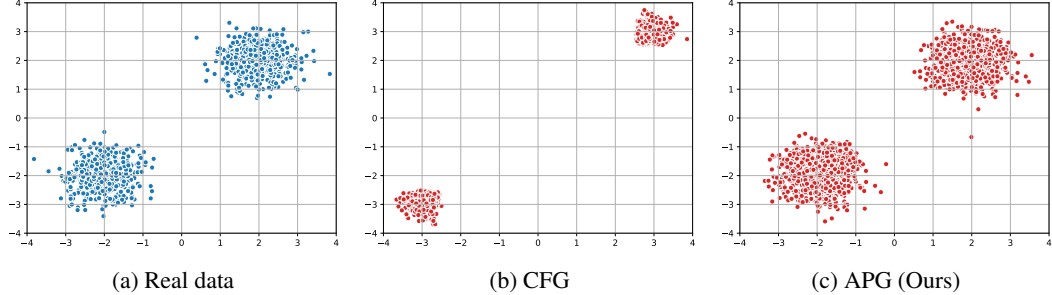

|                | (a) Real data | (b) CFG | (c) APG (Ours) |

Figure 13: Visualizing the effect of APG on the sampling process using a toy problem. The real samples from the data distribution are shown in (a). When sampling with high guidance, CFG leads to a drift away from the true mean of the data distribution and results in reduced mode coverage in the generated samples (b). In contrast, sampling with APG eliminates the drift and increases the coverage of the distribution (c). We used the EDM sampler (Karras et al., 2022) for this experiment.

Table 4: Compatibility of APG with CADS (Sadat et al., 2024a) and IG (Kynkäänniemi et al., 2024). Combining APG with other methods that improve diversity results in better FID than each method in isolation.

<table>
<tr><td colspan="6" align="center">(a) CADS</td><td colspan="6" align="center">(b) Interval guidance (IG)</td></tr>
<tr><td>Guidance</td><td>FID ↓</td><td>Precision ↑</td><td>Recall ↑</td><td>Saturation ↓</td><td>Contrast ↓</td><td>Guidance</td><td>FID ↓</td><td>Precision ↑</td><td>Recall ↑</td><td>Saturation ↓</td><td>Contrast ↓</td></tr>
<tr><td>CFG</td><td>10.42</td><td>0.85</td><td>0.48</td><td>0.46</td><td>0.27</td><td>CFG</td><td>10.42</td><td>0.85</td><td>0.48</td><td>0.46</td><td>0.27</td></tr>
<tr><td>+CADS</td><td>8.65</td><td>0.85</td><td>0.56</td><td>0.43</td><td>0.26</td><td>+IG</td><td>7.49</td><td>0.84</td><td>0.60</td><td>0.39</td><td>0.25</td></tr>
<tr><td>+APG</td><td>6.49</td><td>0.85</td><td>0.62</td><td>0.33</td><td>0.21</td><td>+APG</td><td>6.49</td><td>0.85</td><td>0.62</td><td>0.33</td><td>0.21</td></tr>
<tr><td>+both</td><td>5.56</td><td>0.84</td><td>0.64</td><td>0.32</td><td>0.21</td><td>+both</td><td>5.29</td><td>0.84</td><td>0.65</td><td>0.33</td><td>0.22</td></tr>
</table>

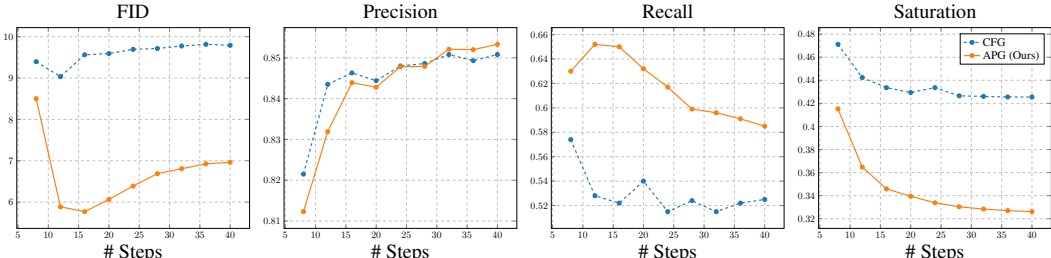

Figure 14: Comparison of CFG and APG across different numbers of sampling steps. APG consistently achieves better FID and recall while maintaining comparable or superior precision to CFG.

## C.2 APG VS NUMBER OF SAMPLING STEPS

We now present the performance comparison between APG and CFG across different numbers of sampling steps using the EDM2 model. Figure 14 indicates that APG consistently provides better FID, recall, and saturation while maintaining the same level of precision.

## C.3 USING APG WITH DISTILLED MODELS

In this section, we show the compatibility of APG with distilled models using PIXART-δ (Chen et al., 2024), SDXL-Lightning (Lin et al., 2024c), and SDXL-Flash (sd community). Consistent with the main text, Figure 15 demonstrates that replacing CFG with APG significantly improves generation quality and saturation level across all models. This is also consistent with Figure 14, where APG outperforms CFG at fewer sampling steps (e.g., 8-16).

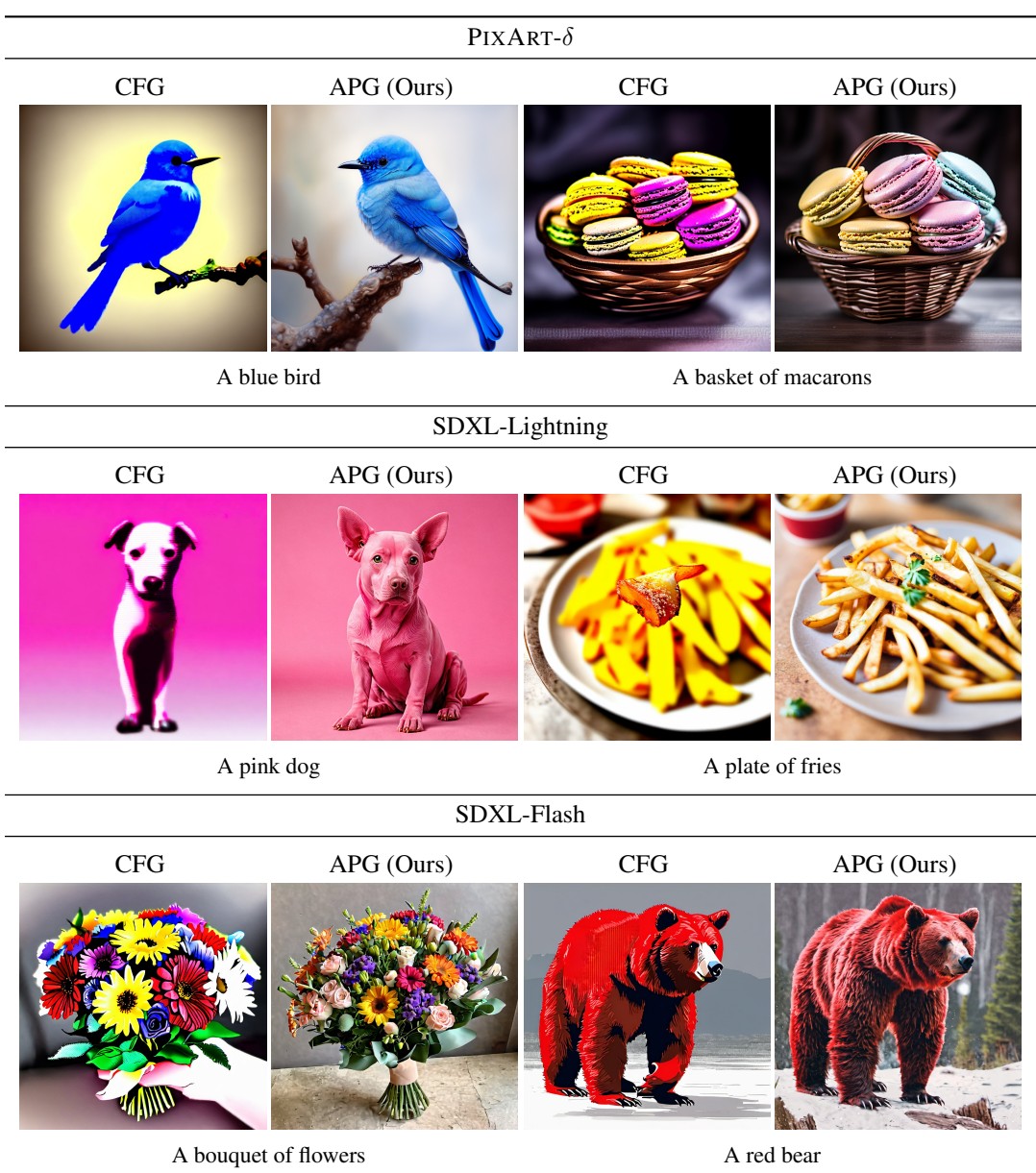

Figure 15: Extended results showcasing the compatibility of APG with distilled diffusion models. Compared to CFG, using APG does not lead to degradation in the output quality.

## C.4    COMPATIBILITY WITH DIFFERENT SAMPLERS

While the main experiment in Table 1 used the default sampler of each model, we next separately show that APG is compatible with different sampling algorithms widely used with diffusion models. As shown in Table 5, using APG with different samplers results in improved FID, recall, and saturation scores, consistent with the main findings in Table 1. We used class-conditional generation using DiT-XL/2 for this experiment.

## C.5    COMPATIBILITY WITH ICG

Independent condition guidance (ICG) (Sadat et al., 2024b) is a method to apply CFG without the need to query an unconditional model. In Table 6, we show that APG is compatible with ICG, and

Table 5: Impact of using APG with popular diffusion samplers using the class-conditional ImageNet model (DiT-XL/2). Compared to CFG, APG showes improved metrics across all samplers.

| Sampler | APG (Ours) | | | CFG | | |
|---|---|---|---|---|---|---|
| | FID ↓ | Recall ↓ | Saturation ↓ | FID ↓ | Recall ↓ | Saturation ↓ |
| DDIM (Song et al., 2021a) | **6.69** | **0.62** | **0.30** | 17.45 | 0.38 | 0.42 |
| DPM++ (Lu et al., 2022b) | **6.87** | **0.62** | **0.32** | 17.65 | 0.38 | 0.43 |
| SDE-DPM++ (Lu et al., 2022b) | **8.53** | **0.57** | **0.32** | 19.01 | 0.36 | 0.43 |
| PNDM (Liu et al., 2022a) | **5.37** | **0.68** | **0.32** | 16.50 | 0.40 | 0.43 |
| UniPC (Zhao et al., 2023) | **6.91** | **0.62** | **0.32** | 17.65 | 0.38 | 0.43 |

Table 6: Compatibility of APG and ICG. Combining APG with ICG significantly improves FID, recall, and saturation scores while maintaining similar precision.

| Guidance | FID ↓ | Precision ↑ | Recall ↑ | Saturation ↓ | Contrast ↓ |
|---|---|---|---|---|---|
| ICG | 17.63 | **0.85** | 0.32 | 0.49 | 0.28 |
| +APG (Ours) | **5.73** | **0.85** | **0.63** | **0.33** | **0.22** |

Table 7: Compatibility of APG and TSG with. Combining APG with TSG improves FID, recall, and saturation scores while maintaining similar precision.

| Guidance | FID ↓ | Precision ↑ | Recall ↑ | Saturation ↓ | Contrast ↓ |
|---|---|---|---|---|---|
| TSG | 14.00 | **0.81** | 0.52 | 0.37 | 0.28 |
| +APG (Ours) | **5.84** | **0.81** | **0.66** | **0.30** | **0.20** |

similar to CFG, using APG with ICG results in improved FID, recall, and saturation scores while maintaining similar precision. We use class-conditional ImageNet generation with EDM2-S for this experiment.

## C.6 COMPATIBILITY WITH TSG

Time-step guidance (TSG) (Sadat et al., 2024b) is an extension of CFG that leverages the time-step information learned by the diffusion model to enhance the quality of generations. We next demonstrate that applying the update rule in APG further improves the performance of TSG. Table 7 shows that APG improves FID, recall, and saturation metrics, while maintaining similar precision to TSG. This experiment is based on class-conditional ImageNet generation using DiT-XL/2.

## C.7 ALIGNMENT WITH THE CONDITION

We next demonstrate that replacing CFG with APG does not compromise the alignment between the input condition and the output. To validate this, we measure the classification accuracy of the generated results for the ImageNet task and the CLIP score for Stable Diffusion. The results in Table 8 show that both CFG and APG achieve comparable alignment metrics. Thus, APG reduces saturation and improves FID without compromising condition alignment.

Table 8: Condition alignment comparison between CFG and APG.

| Alignment metric | CFG | APG |
|---|---|---|
| Class Accuracy ↑ | 0.97 | 0.96 |
| CLIP-Score ↑ | 0.31 | 0.31 |

## C.8 EXTENDED ABLATION STUDIES

**Effect of the parallel component** We next demonstrate the effect of $\eta$ on the generated images in Table 9a. As hypothesized in Section 4, increasing the strength of the parallel component leads to higher saturation levels and increased FID. We recommend setting $\eta = 0$ by default and only increasing it if more saturation is desired in the generated images.

Table 9: Ablation study examining various design elements in APG.

| (a) Influence of $\eta$ | | | | (b) Impact of rescaling $r$ | | | | (c) Effect of momentum $\beta$ | | |
|---|---|---|---|---|---|---|---|---|---|---|
| $\eta$ | FID $\downarrow$ | Saturation $\downarrow$ | | $r$ | FID $\downarrow$ | Recall $\uparrow$ | | $\beta$ | FID $\downarrow$ | Recall $\uparrow$ |
| 0.0 | **6.49** | **0.33** | | 0.25 | 7.45 | **0.72** | | $-1.5$ | 13.38 | **0.73** |
| 0.25 | **6.49** | 0.34 | | 2.5 | **6.49** | 0.62 | | $-0.75$ | **6.49** | 0.62 |
| 0.5 | **6.49** | 0.36 | | 10 | 7.97 | 0.57 | | 0.0 | 6.84 | 0.60 |
| 1.0 | 6.63 | 0.37 | | $\infty$ | 7.93 | 0.56 | | 0.5 | 7.10 | 0.59 |

Table 10: Hyperparameters used in the main experiment (Table 1).

| Model | $w$ | $\eta$ | $r$ | $\beta$ |
|---|---|---|---|---|
| EDM2-S | 4 | 0 | 2.5 | $-0.75$ |
| EDM2-XL | 2 | 0 | 2.5 | $-0.75$ |
| DiT-XL/2 | 4 | 0 | 5 | $-0.50$ |
| Stable Diffusion 2.1 | 10 | 0 | 7.5 | $-0.75$ |
| Stable Diffusion XL | 15 | 0 | 15 | $-0.50$ |

**Effect of the rescaling threshold**     The effect of the rescaling radius $r$ on the generated images is shown in Table 9b. Excessive rescaling degrades image quality, while high values of $r$ result in no noticeable change, as the rescaling function approaches the identity function. Therefore, midrange values for $r$ yield better FID scores. We suggest observing the norm of $\Delta D_t$ during the inference process and choosing $r$ in a way that is comparable (on average) to the norm of $\Delta D_t$.

**Effect of the momentum strength**     Table 9c shows the effect of momentum strength $\beta$ on generation quality. Note Negative values for $\beta$ result in better FID compared to positive momentum, and excessive momentum degrades image quality. This aligns with our hypothesis that moving away from the previous directions helps limit the drift that can occur during sampling with higher guidance scales. Empirically, we found that $\beta \in [-0.75, -0.25]$ works well in most setups.

## D    IMPLEMENTATION DETAILS

We provide the code for APG in Algorithm 1. Compared to CFG, APG only includes a few additional lines of code without noticeable computational overhead. As discussed in Section 4, we always convert the predictions of the diffusion model to $D_{\boldsymbol{\theta}}(\boldsymbol{z}_t, t, \boldsymbol{y})$, compute the guided prediction, and convert it back to the initial output type at each sampling step.

We mainly use the ADM evaluation suite (Dhariwal & Nichol, 2021) for computing FID, precision, and recall. The FID is computed using 10,000 generated images and the whole training set for class-conditional ImageNet models. For text-to-image models, the FID is evaluated using the evaluation subset of MS COCO 2017 (Lin et al., 2014). The hyperparameters used for the main experiment are given in Table 10.

## E    MORE VISUAL RESULTS

This section presents extended visual comparisons between APG and CFG. Additional results using EDM2 are provided in Figure 16, with an example of how APG enhances diversity shown in Figure 17. Further images for Stable Diffusion 2.1 and Stable Diffusion XL are included in Figures 18 and 19. Moreover, Figure 20 illustrates how APG improves text spelling in Stable Diffusion 3. Finally, more examples of APG applied to distilled models are shown in Figures 21 to 23.

**Algorithm 1** PyTorch implementation of APG.

```python
import torch

class MomentumBuffer:
    def __init__(self, momentum: float):
        self.momentum = momentum
        self.running_average = 0

    def update(self, update_value: torch.Tensor):
        new_average = self.momentum * self.running_average
        self.running_average = update_value + new_average

def project(
    v0: torch.Tensor, # [B, C, H, W]
    v1: torch.Tensor, # [B, C, H, W]
):
    dtype = v0.dtype
    v0, v1 = v0.double(), v1.double()
    v1 = torch.nn.functional.normalize(v1, dim=[-1, -2, -3])
    v0_parallel = (v0 * v1).sum(dim=[-1, -2, -3], keepdim=True) * v1
    v0_orthogonal = v0 - v0_parallel
    return v0_parallel.to(dtype), v0_orthogonal.to(dtype)

def adaptive_projected_guidance(
    pred_cond: torch.Tensor,   # [B, C, H, W]
    pred_uncond: torch.Tensor, # [B, C, H, W]
    guidance_scale: float,
    momentum_buffer: MomentumBuffer = None,
    eta: float = 1.0,
    norm_threshold: float = 0.0,
):
    diff = pred_cond - pred_uncond
    if momentum_buffer is not None:
        momentum_buffer.update(diff)
        diff = momentum_buffer.running_average
    if norm_threshold > 0:
        ones = torch.ones_like(diff)
        diff_norm = diff.norm(p=2, dim=[-1, -2, -3], keepdim=True)
        scale_factor = torch.minimum(ones, norm_threshold / diff_norm)
        diff = diff * scale_factor
    diff_parallel, diff_orthogonal = project(diff, pred_cond)
    normalized_update = diff_orthogonal + eta * diff_parallel
    pred_guided = pred_cond + (guidance_scale - 1) * normalized_update
    return pred_guided
```

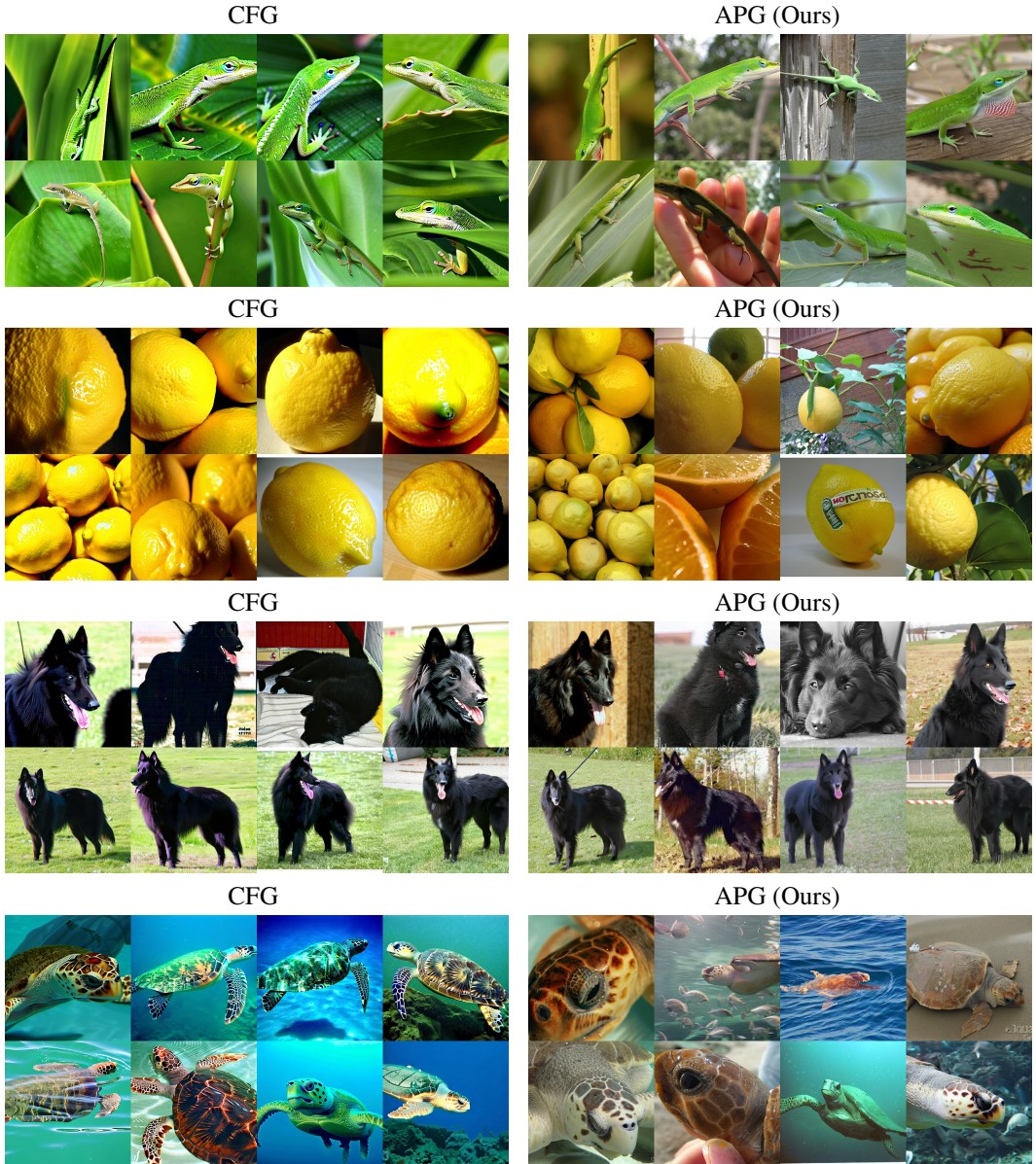

Figure 16: More visual results comparing APG and CFG using EDM2.

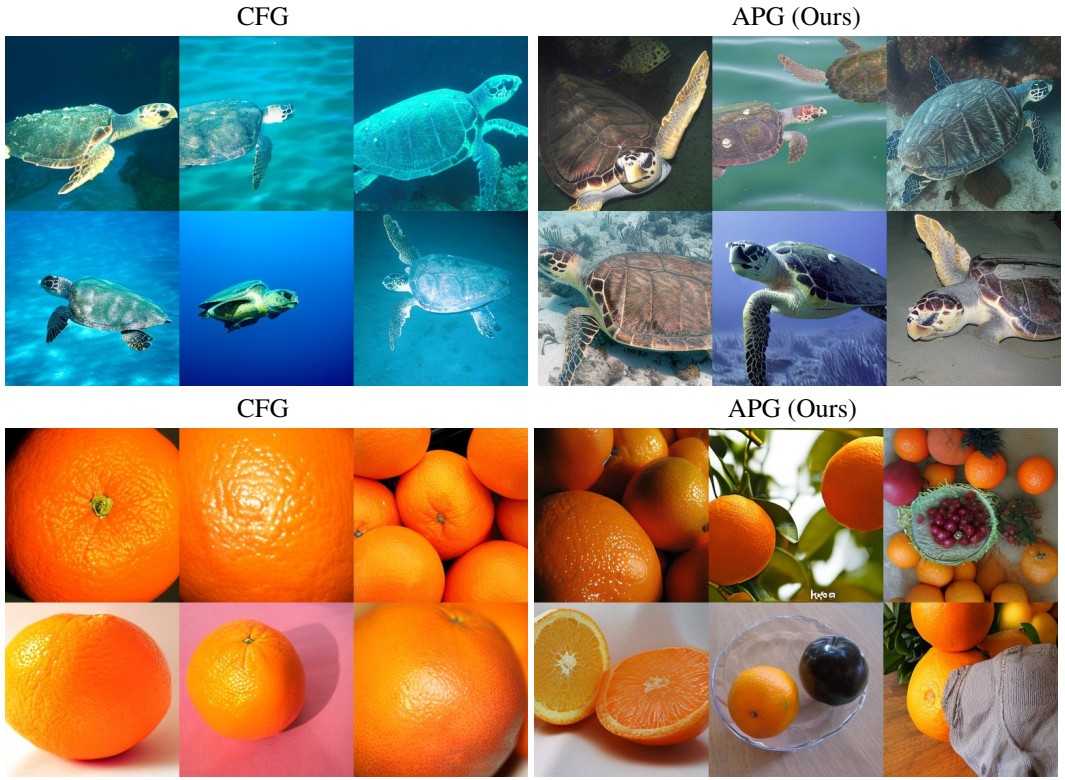

CFG      APG (Ours)

CFG      APG (Ours)

Figure 17: Showcasing the diversity of generations after using APG. APG removes oversaturation issues while improving diversity w.r.t. the overall image composition.

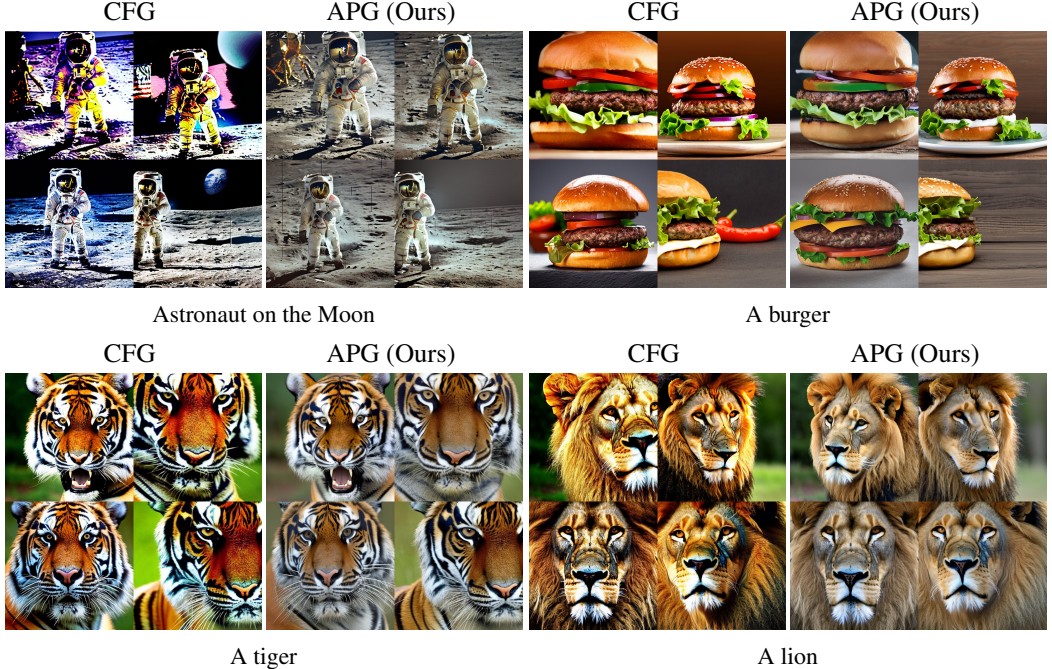

CFG  APG (Ours)  CFG  APG (Ours)

Astronaut on the Moon    A burger

CFG  APG (Ours)  CFG  APG (Ours)

A tiger     A lion

Figure 18: More visual examples comparing CFG and APG using Stable Diffusion 2.1.

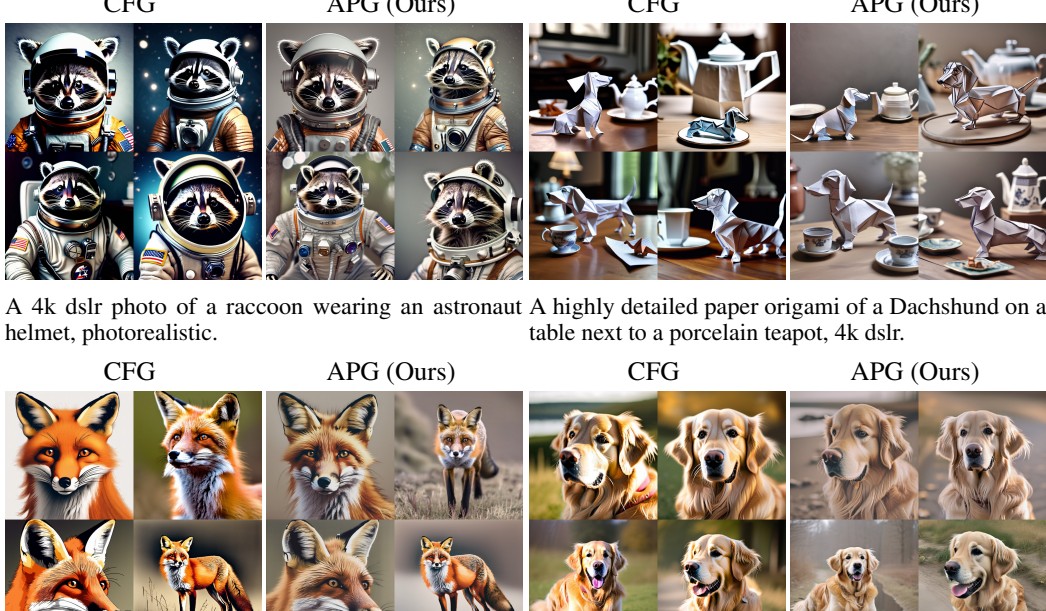

Figure 19: More visual examples comparing CFG and APG using Stable Diffusion XL.

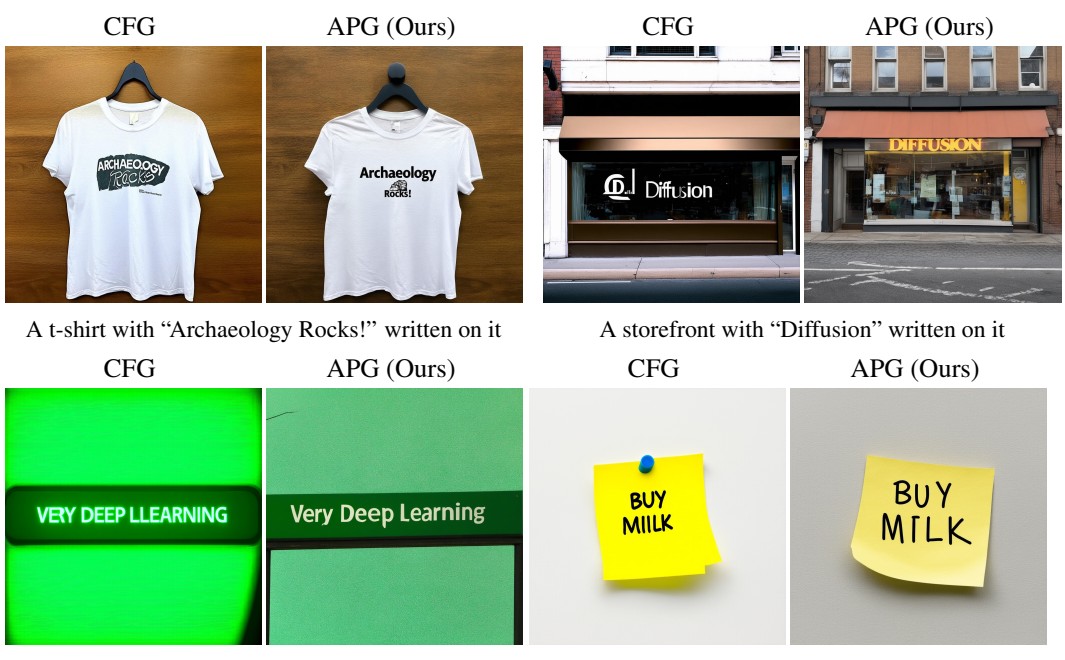

Figure 20: Additional visual examples on the quality of rendering text using Stable Diffusion 3. Compared to CFG, APG results show more consistent spellings.

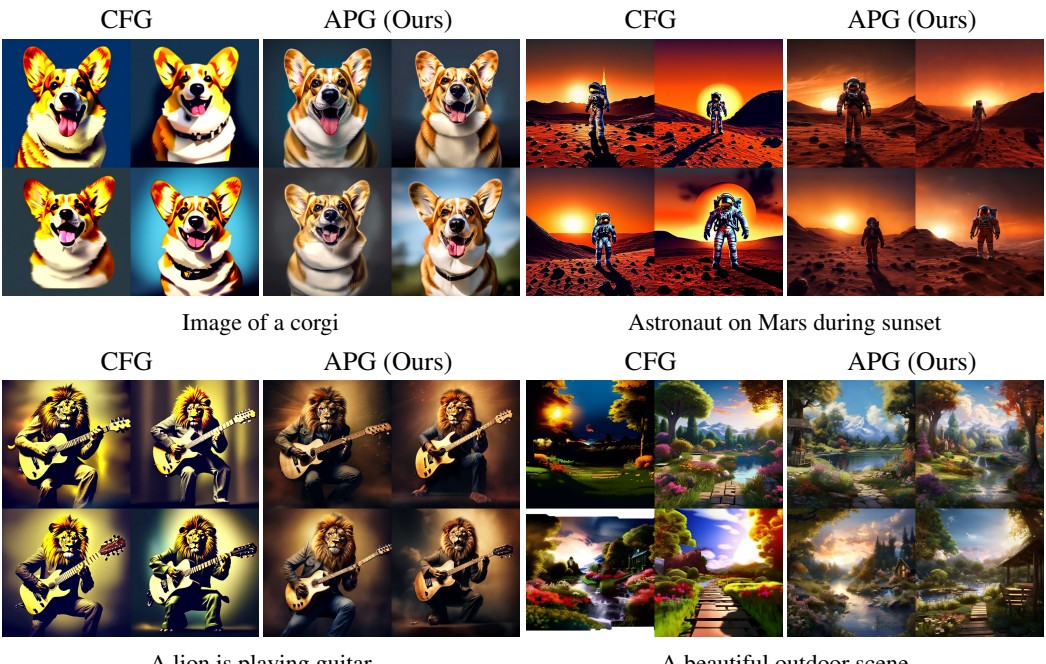

Figure 21: More visual examples comparing CFG and APG using PIXART-$\delta$.

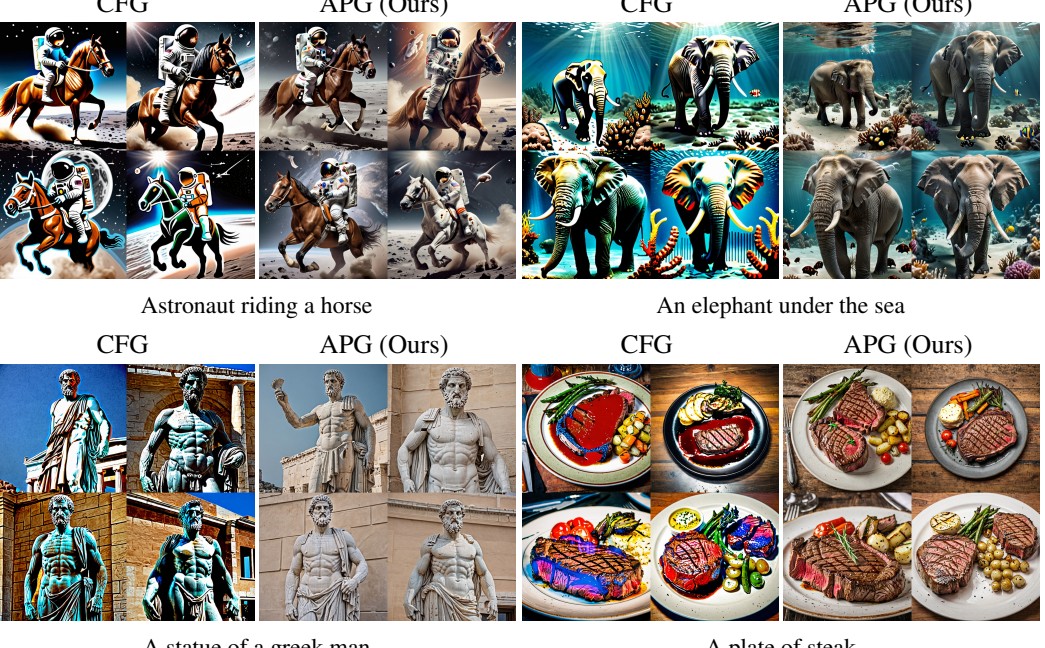

Figure 22: More visual examples comparing CFG and APG using SDXL-Flash.

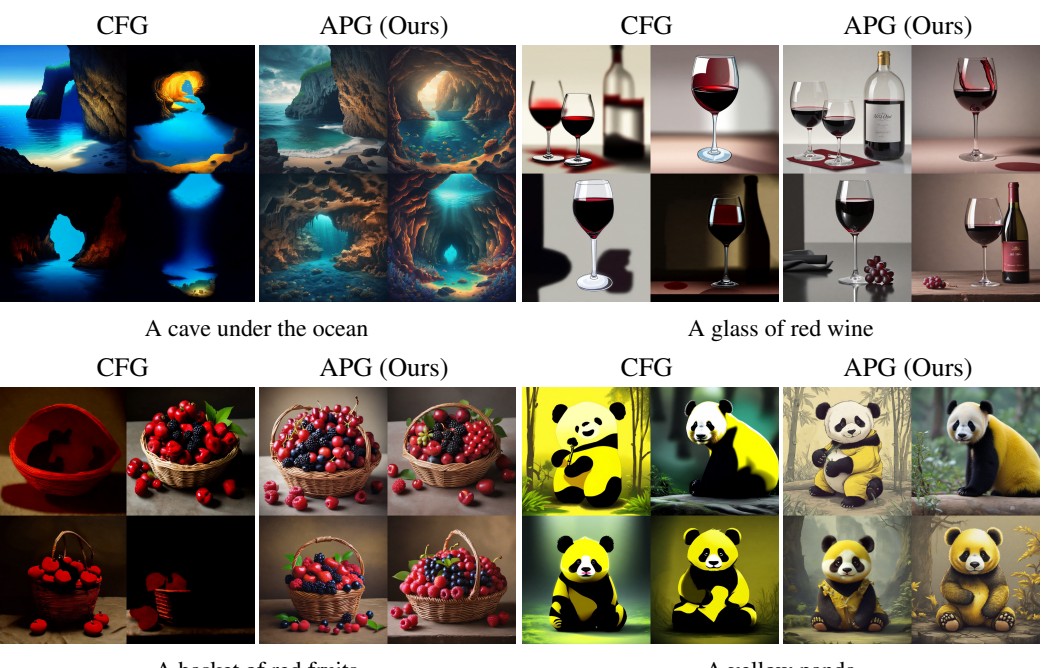

Figure 23: More visual examples comparing CFG and APG using SDXL-Lightning.

