# OpenReview forum: "Eliminating Oversaturation and Artifacts of High Guidance Scales in Diffusion Models"
_ICLR.cc/2025/Conference — ICLR 2025 Poster_

### Official Review · Reviewer_LhNz · 2024-10-26

**Soundness:** 3
**Presentation:** 4
**Contribution:** 3
**Rating:** 6
**Confidence:** 3

**Summary:**

This paper presents a method called Adaptive Projected Guidance (APG) to address the issues of oversaturation and artifacts that occur when using high guidance scales in Classifier-Free Guidance (CFG) for diffusion models. The authors decompose the CFG update rule into two components: a parallel component responsible for oversaturation and an orthogonal component that enhances image quality. They propose down-weighting the parallel component to reduce oversaturation. Additionally, they introduce rescaling and reverse momentum to improve sampling. Extensive experiments demonstrate that APG can improve FID, recall, and saturation scores while maintaining precision, making it a superior alternative to standard CFG.

**Strengths:**

1. APG is presented as a plug-and-play method with virtually no additional computational overhead, making it attractive for practical use in various diffusion models.
2. The method of down-weighting the parallel component in CFG is novel and interesting, effectively reducing oversaturation and preserving image quality at high guidance scales.

**Weaknesses:**

1. In Figure 8, could the authors explain why the proposed APG shows lower precision and higher FID compared to CFG when the guidance scale  w  is below certain values (e.g.,  w = 2.5 , where APG precision is lower than CFG)?
2. While reverse momentum is introduced, the intuition behind why it improves the image generation quality could be expanded. More detailed theoretical or experimental exploration of this component would clarify its importance.

**Questions:**

See the questions in the weakness section.

---

> ### Author Response · Authors · 2024-11-15
> **Official Response to Reviewer LhNz**
>
> We greatly appreciate the reviewer's helpful suggestions, as well as the positive assessment of the influence and quality of our work. Below, we provide detailed responses to the reviewer’s comments.
>
> ### **Lower precision at lower guidance scales**
> The results in Figure 8 were compiled using fixed hyperparameters for momentum and normalization, which suggests that slight adjustments to these hyperparameters may be necessary depending on the guidance scale. At very low scales, the original CFG does not exhibit oversaturation, whereas APG with the fixed parameters from Figure 8 may introduce excessive normalization. However, as the guidance scale increases, the precision and FID of CFG degrade, while APG consistently maintains its quality. This is reflected in APG achieving better FID across a range of guidance scales and a superior minimum FID overall. Thus, APG enables the use of higher guidance scales without encountering the typical issues observed with CFG. Additionally, in our experiments, we found that by reducing normalization and momentum for lower guidance scales, APG can achieve results similar to or better than CFG.
>
> ### **Motivation behind momentum**
> Our motivation for incorporating momentum is to filter out any common signals present across different CFG updates at various time steps. By taking a weighted running average of past updates and removing the overlapping component from the current update, we effectively retain only the new information at each step. We believe this approach helps limit the drift typically seen in CFG over multiple sampling steps. Table 2 shows that removing momentum results in a higher FID, demonstrating that momentum contributes to improved quality. We would be happy to expand on the intuition or add additional experiments on momentum in the final version.

---

> > ### Comment · Reviewer_LhNz · 2024-11-27
> >
> > Thank you for the response. I will keep my rating.

---

### Official Review · Reviewer_mQei · 2024-10-27

**Soundness:** 4
**Presentation:** 4
**Contribution:** 3
**Rating:** 6
**Confidence:** 3

**Summary:**

The paper introduces a novel method called Adaptive Projected Guidance (APG) to address the issue of oversaturation and unrealistic artifacts that occur when using high guidance scales in diffusion models. Specifically, the authors propose a modification to the classifier-free guidance (CFG) update rule by decomposing the update term into parallel and orthogonal components relative to the conditional model prediction. The authors conduct extensive experiments to support their claims.

**Strengths:**

1. The paper introduces a new method, APG, that addresses a significant problem in diffusion models—oversaturation and artifacts associated with high guidance scales. The approach is innovative and directly tackles a well-known issue in the field.
2. The authors provide extensive experimental results to validate their claims. They demonstrate the effectiveness of APG across various models and samplers, showing improvements in FID, recall, and saturation scores.
3. The paper provides a clear comparison between APG and CFG, both qualitatively and quantitatively. The visual results are particularly compelling in showing the reduction in oversaturation and artifacts.

**Weaknesses:**

1. The final example generated by APG in Figure 10 has a logical error, where the cat has only one paw.
2. For Table 1, the authors should consider adding HPSv2 and ImageReward metrics to evaluate the generation performance.
3. For Table 1, why is the guidance scale of Stable Diffusion XL so large (w=15)? When reducing the guidance scale, are the performance improvements limited? The authors should provide more quantitative results.
4. The authors claim that the orthogonal component is chiefly responsible for improvements in image quality, while the parallel component increases saturation in the generations. How many examples do the authors observe? Are there any theories to support this observation?

**Questions:**

Please refer to the Weakness section.

---

> ### Author Response · Authors · 2024-11-15
> **Official Response to Reviewer mQei**
>
> We wish to thank the reviewer for the helpful comments and for finding our work novel with detailed evaluations, good presentation, and significant contribution. Please find our answers to the comments below.
>
> ### **HPSv2 and ImageReward**
> We will examine the suggested metrics and will gladly consider them for the final version of the paper.
>
> ### **Note on lower guidance scales**
> Lower guidance scales generally avoid oversaturation, but often compromise output quality and alignment with the input. Throughout our experiments, we observed that APG performs robustly across a wide range of guidance scales, as illustrated in Figure 8. Additionally, we demonstrated that APG enables the use of higher guidance scales (such as $w=15$ for Stable Diffusion XL) that are typically infeasible with CFG, achieving high-quality generations that align well with the input condition without encountering common CFG issues. Thus, we do not believe that the benefits of APG are limited to certain guidance scales shown in the paper.
>
> ### **Effect of the parallel component**
> As shown in Table 2, results with and without the parallel component yield similar FID scores (computed over 10,000 generated samples); however, the model with the parallel component exhibits higher saturation. This suggests that the orthogonal component primarily drives FID, while the parallel component mainly influences saturation. For theoretical insight into this effect, we refer the reviewer to our “gain” argument in Section 4 (beginning around line 186 in the original manuscript).
>
> ### **Example in figure 10**
> This figure primarily illustrates improvements in text quality, where "hello world" is misspelled in the CFG figure. Since the base image for CFG contains artifacts (including two extra paws around the sign), our method successfully resolves one but not all of these artifacts in this case.

---

### Official Review · Reviewer_xLYc · 2024-10-30

**Soundness:** 3
**Presentation:** 3
**Contribution:** 3
**Rating:** 6
**Confidence:** 2

**Summary:**

This work aims to reduce the oversaturation in the current CFG approach. The authors decompose the CFG update term into two components—parallel and orthogonal—and both theoretically and empirically demonstrate the role of each component. Based on these observations, they redesign the weighting of each component and additionally propose an adaptive rescaling strategy.

**Strengths:**

- The intuition and theory are well-balanced.
- The results are quite promising. Their guidance does not drastically alter the image content, which aligns well with their claims.
- They demonstrate robustness across various types of diffusion models.

**Weaknesses:**

- I do not view this as a straightforward rescaling of CFG. However, it might be open to interpretation, and I would be interested to hear the authors’ perspective on this.

- Which model was used for Figure 8?
- Further analysis on the rescale weight would be helpful—does it show any trends with respect to t?

- *(As a suggestion)* Including illustrations of the geometry, along with the related intuitions, would further enhance the clarity.

**Questions:**

Please refer to the weaknesses section.

---

> ### Author Response · Authors · 2024-11-15
> **Official Response to Reviewer xLYc**
>
> We thank the reviewer for the positive reception of our paper and for recognizing its strengths. We also thank the reviewer for the opportunity to clarify several points.
>
>
> ### **Analysis on rescaling the guidance weight**
> We believe that the impact of APG extends beyond merely adjusting the guidance weight. A key observation is that modifying the parallel component causes a deviation from the CFG update rule. If we express CFG as $D(x, t, y) + w(D(x, t, y) - D(x, t))$, then we can show that applying the projection alters the update to $w’ D(x, t, y) + w (D(x, t, y) - D(x, t))$ for some constant $w’ \neq 1$. This transformation cannot be achieved by simply modifying the guidance scale $w$ alone. Therefore, we argue that APG achieves more than just rescaling the CFG weight.
>
> Furthermore, in Appendix C.1, we present experiments combining APG with other methods that effectively adjust the guidance weight, specifically CADS and Interval Guidance (IG). The results indicate that combining APG with these methods yields the best FID compared to each method in isolation. This suggests that APG’s benefits are, to some extent, orthogonal to CFG weight adjustments.
>
> ### **Model for figure 8**
> Figure 8 was generated using the EDM2 model for class-conditional ImageNet generation.
>
> ### **The trend of the rescale weight**
> We observed that the norm of the updates increases during the initial steps of inference and then gradually approaches zero as denoising progresses. However, identifying a consistent pattern for this behavior is challenging, as it dynamically varies depending on the input prompt and random seed. We are happy to provide further analysis if this does not fully address the reviewer's comment.
>
> ### **Illustrations of the geometry**
> We would be glad to provide geometric diagrams illustrating the visual interpretation of each step in APG to help guide the reader’s intuition for our approach. If this does not fully address the reviewer's point, we welcome further discussion.

---

> > ### Comment · Reviewer_xLYc · 2024-11-21
> >
> > Regarding Q2, it seems that there is a discrepancy between the minimum point in Figure 8 and the values in Table 1. Were different configurations used?
> >
> > Also, I’m not familiar with EDM2—does the original paper suggest an ideal guidance value? From what I’ve seen in the paper, values not exceeding 2 seem to be used. If there’s a range difference between those values and the ones you used (e.g., w=2, 4), was APG what made it possible?

---

> ### Author Response · Authors · 2024-11-21
> **Official Response to Reviewer xLYc**
>
> Table 1 aims to illustrate the differences between APG and CFG at higher guidance scales, with results reported using $w = 4$. This choice is based on the observation that while lower guidance scales yield better FID values, higher guidance scales produce better visual quality as judged by human inspection. However, at these higher guidance scales (e.g., $w = 4$), standard CFG starts exhibiting saturation and artifacts. APG effectively addresses these issues, resulting in improved metrics. In Figure 8, we separately show that the superior metrics achieved by APG are not restricted to a specific guidance weight, i.e., APG outperforms CFG across a range of guidance values.
>
>
> For EDM2 [1], the paper does not recommend a specific scale but uses $w = 1.2 -1.4$ for their final models. We have observed that above this level, the saturation and artifacts begin to become more pronounced with the standard CFG. APG is designed to easily allow scales above this level (e.g., $w = 4$) while avoiding these issues.
>
> [1] Karras, Tero, Miika Aittala, Jaakko Lehtinen, Janne Hellsten, Timo Aila, and Samuli Laine. "Analyzing and improving the training dynamics of diffusion models." In Proceedings of the IEEE/CVF Conference on Computer Vision and Pattern Recognition, pp. 24174-24184. 2024.

---

> > ### Comment · Reviewer_xLYc · 2024-11-26
> >
> > Thank you for your kind response. My questions have been thoroughly addressed, and I will keep my score as it is.

---

### Author Response · Authors · 2024-11-24
**Message to All Reviewers**

We thank the reviewers once again for their positive assessment of our paper’s contribution and presentation and for their helpful comments. We hope that our replies have satisfactorily addressed the reviewers’ questions and further strengthened their assessment of our work. We remain open to discussion should any issues remain.

---

### Meta-Review · Area_Chair_jCq1 · 2024-12-22

**Metareview:**

This work presents a novel method called Adaptive Projected Guidance (APG) to address the issue of oversaturation and unrealistic artifacts that occur when using high guidance scales in diffusion models. To do so, the authors propose a modification to the classifier-free guidance (CFG) update rule by decomposing the update term into parallel and orthogonal components relative to the conditional model prediction. Experimental results have verified the superior performance of the developed method.

**Additional Comments On Reviewer Discussion:**

This work has three reviewers. All three reviewers are positive to accept this work, and all ratings are 6. And the reviewers also claimed that the rebuttal well addressed their issues during the rebuttal stage. In this regards, this work can be accepted in ICLR 2025.

---

### Decision · Program_Chairs · 2025-01-22

Accept (Poster)